



# Cosmogenic $^3$He dating of olivine with tightly retained mantle $^3$He, Volcano Mountain, Yukon

Jessica M. Mueller[1], Jeff D. Bond[2], Kenneth A. Farley[1], Brent C. Ward[3]

[1]Division of Geological & Planetary Sciences, California Institute of Technology, Pasadena, 91125, USA
[2]Yukon Geological Survey, Yukon, Y1A 2C6, Canada
[3]Simon Fraser University, Burnaby, V5A 1S6, Canada

*Correspondence to*: Jessica M. Mueller (jessica@caltech.edu)

**Abstract.** We present a step-heat method for isolating cosmogenic $^3$He ($^3$He$_c$) from mantle He in olivine xenocrysts to date the eruption of very young nephelinites from Volcano Mountain (VM) Yukon, Canada. In these olivines, the standard
procedure of powdering grains to <30 μm failed to adequately remove mantle helium prior to fusion analyses. For example, in one powder fusion the concentration of $^4$He was 2.93 x $10^6$ ± 6.04 x $10^4$ Matoms/g with a $^3$He/$^4$He ratio of 8.7 ± 0.3 R$_A$ (atmospheric ratio; R$_A$ = 1.384 x $10^{-6}$). Based on the $^3$He/$^4$He ratio of 8.1 ± 0.2 R$_A$ released by crushing of the same sample, the estimated fraction of mantle $^3$He in the powder fusion is between 87 % and 98 % of the total $^3$He. The inability to effectively isolate $^3$He$_c$ from these samples likely arises from the survival of small (<<30 μm) fluid inclusions hosting mantle He through
the powdering step. The presence of such unusually small fluid inclusions may relate to the origin of the olivines as disaggregated peridotite xenoliths rather than the more commonly analyzed olivine phenocrysts. Regardless, the high proportion of mantle $^3$He in the powder fusion yields highly uncertain $^3$He$_c$ exposure ages. We circumvented this problem by heating powdered olivine in a three-step heating schedule ranging from 700 to 1400 °C. 80-92 % of $^3$He$_c$ was released in the low temperature step and the rest was released in the middle temperature step. By the highest temperature step, the released
He had a mantle-like $^3$He/$^4$He ratio. Using this technique on two samples from the youngest VM flow, we obtained precise estimates of cosmogenic $^3$He concentrations, from which we derive an eruption age of 10.9 ka ± 1.1 ka.

## 1 Introduction

The in-situ production of cosmogenic $^3$He in olivine has been used to date the surface exposures of lava flows for decades (e.g.
Aciego et al., 2007; Kurz et al., 1990; Marchetti et al., 2020). The abundance of mm-size olivine crystals in basalt and the high helium retentivity of olivine (Shuster and Farley, 2004; Trull and Kurz, 1993) make it an ideal candidate for cosmogenic $^3$He dating. The method is particularly suited to determine the exposure ages of young volcanics (Heineke et al., 2016; Fenton and Niedermann, 2014; Licciardi et al., 1999) because $^3$He has one of the highest production/detection limit ratios (Niedermann, 2002).

Cosmogenic $^3$He in terrestrial rocks is primarily produced by spallation of target nuclei in the crystal lattice of minerals. At surface temperatures, $^3$He$_c$ accumulates quantitatively in the matrix of olivine (Kurz, 1986a, b), allowing exposure ages to





be obtained from independently established production rates. Unlike cosmogenic radionuclides such as $^{10}$Be and $^{26}$Al in which the background nuclide concentration is often negligible, a practical challenge of the $^3$He method is isolating the cosmogenic component from other $^3$He components. The most common non-cosmogenic sources of $^3$He include thermal neutron capture

on $^6$Li (Andrews and Kay, 1982; nucleogenic $^3$He, $^3$He$_{nuc}$) and mantle He with $^3$He/$^4$He ratios well above crustal values trapped in fluid inclusions (Kurz, 1986a, b). U and Th-bearing minerals also produce radiogenic $^4$He ($^4$He$_{rad}$) through α decay. Taken together:

$$^3\text{He}_{total} = {}^3\text{He}_c + {}^3\text{He}_m + {}^3\text{He}_{nuc} \tag{1}$$

$$^4\text{He}_{total} = {}^4\text{He}_m + {}^4\text{He}_{rad} \tag{2}$$

However, concentrations of Li, U, and Th in olivine tend to be low (Woodhead, 1996; Dostal and Capedri, 1975; Kent and Rossman, 2002; Blard and Farley, 2008; Amidon et al., 2009; Blard et al., 2013) such that neutron capture and radioactive decay produces minimal amounts of He in most young lava flows. Thus, in this specific case:

$$^3\text{He}_{total} = {}^3\text{He}_c + {}^3\text{He}_m \tag{3}$$

$$^4\text{He}_{total} = {}^4\text{He}_m \tag{4}$$

Equation 3 highlights that cosmogenic $^3$He dating in young olivine requires the isolation of $^3$He$_c$ from $^3$He$_m$. Past studies (Kurz, 1986a) achieved sufficient separation of the two components using a two-step process in which the first step is crushing whole olivine grains in vacuum and measuring the $^3$He/$^4$He ratio. Vacuum-crushing olivine grains to <500 μm releases most mantle He in fluid inclusions and yields the crush ratio for a given sample ($^3$He/$^4$He)$_{crush}$ (Kurz, 1986a, 1986b; Puchol et al., 2017; Blard, 2021). In the second step, the crushate is powdered to eliminate any surviving fluid inclusions, and then fused in vacuum

to measure matrix hosted He. Crush ratios (($^3$He/$^4$He)$_{crush}$) are often close to the upper mantle ratio of ~8 times the atmospheric ratio (1.384 x 10$^{-6}$, R$_A$) in basalts while fusion ratios ($^3$He/$^4$He)$_{fusion}$) of cosmic-ray irradiated samples may be much higher (Kurz, 1986a, b). If $^4$He$_{fusion}$ is assumed to be entirely mantle derived, then the cosmogenic component can be isolated even if some of the He in the fusion is mantle derived:

$$^3\text{He}_c = {}^3\text{He}_{fusion} - ({}^4\text{He}_{fusion})({}^3\text{He}/{}^4\text{He})_{crush} \tag{5}$$

$^3$He$_c$ is then used to calculate an exposure age. The uncertainty on $^3$He$_c$ depends in part on how large and how well-known the correction for mantle He is. When powdering does not effectively remove the mantle component, the correction can be large compared to the total $^3$He measurement. An alternative way to estimate the cosmogenic component is to create a helium isochron (Blard and Pik, 2008; Blard, 2021). In this method, the crush step is skipped, and multiple aliquots (n) of a sample or samples with the same mantle component and exposure age are fused individually and used to create a helium isochron based

on the following equation:




$$^3He^n_{total} = {}^3He_c + ({}^4He^n_m)({}^3He/{}^4He)_m \qquad (6)$$

For $n \geq 2$, this is a system of equations with 2 unknowns ($^3He_c$, $(^3He/^4He)_m$). If the aliquots have sufficiently different $^4He$ concentrations, a line regressed through a plot of $^3He$ vs $^4He$ has a slope of $(^3He/^4He)_m$ and a y-intercept of $^3He_c$ (Blard, 2021). As with the crush-fusion method, the concentration of $^3He_c$ is used to calculate an exposure age and the uncertainty of this age
rises as $^3He_m$ gets closer to $^3He_{total}$.

In this study, we attempted both the crush-fusion method and the isochron method to obtain exposure ages for the youngest lava flows from Volcano Mountain in Yukon, Canada. In both approaches the mantle component was so large and well-retained that precise $^3He_c$ concentrations could not be obtained. As an alternative we developed a step heating protocol following work on peridotites (Swindle et al., 2023) where step heating samples partially isolated $^3He_c$ from mantle $^3He$.

## 2    Geologic Setting

The Fort Selkirk Volcanic Complex in Yukon, Canada is comprised of valley-filling volcanics that interacted with Cordilleran ice sheets during the Pliocene and Pleistocene epochs (Jackson et al., 2012; Jackson and Huscroft, 2023).Volcano Mountain, located north of the confluence of the Yukon and Pelly Rivers, is the youngest cone in the complex (Jackson and Stevens, 1992). Volcano Mountain rises several hundred meters above the valley fill and is comprised of lava flows that dammed small
lakes in the early-mid Holocene. Radiocarbon ages of lake bottom sediments indicate that the lavas predate 7300-4200 BP (Jackson and Stevens, 1992; Francis & Ludden, 1990).

Volcano mountain lava flows are nephelinites carrying olivine phenocrysts, ultramafic xenoliths, and olivine xenocrysts (Francis & Ludden, 1990; Trupia & Nicholls, 1996). The xenocrysts are large (up to 1 cm) and are the dominant population of olivine crystals in the rock. Based on deformation features such as kink-banding and subgrain boundaries, their
anhedral/broken shape, and highly magnesian composition (Fo88-90), they are likely disaggregated spinel lherzolite xenoliths (Francis & Ludden, 1990; Trupia & Nicholls, 1996). These abundant and large xenocrysts are an attractive target for cosmogenic $^3He$ dating.

### 3. Samples

### 3.1 Field Sampling

To determine the most recent eruption of Volcano Mountain, cosmogenic sampling targeted the youngest flows based on stratigraphy and weathering. (Table 1; Fig.1). Particular attention was paid to the flow within the crater as the most likely to date the last eruption. For redundancy purposes, 2-3 samples were collected at each location. Site parameters used to guide sampling included: a relatively flat, stable surface exhibiting primary depositional features (e.g. ropy textures) and if possible, minimal vegetation cover. A gas-powered cutoff saw was then used to control sampling by slicing the surface to a uniform
depth. A rock hammer and chisel were used to liberate the sample from the prepared area (Fig.1). Approximately 2 kg of





sample was collected from each site. Notes collected included: location, average sample depth, distance from edges, surface slope angle and aspect, weathering characteristics (surface relief), vegetation cover, and topographic shielding.

A total of 13 samples were collected on 6 separate lava flow surfaces at elevations ranging from ~0.7-1.1 km. Jackson and Stevens (1992) mapped VM flows and named them based the direction of the flow (north or south) and its age relative to the

other flows based on field observations (0-2 where 2 is assigned to the youngest flows) (Fig. 1.). Samples VM-01, 02, and 03 were collected from the stratigraphically youngest flow in the cinder cone (Fig. 1; 2Na). VM-06 and VM-07 sample the fifth south flow (Fig. 1; 2Sa). VM-08 and 09 are samples from the third south flow which is the oldest flow based on field observations (Fig. 1; 1Sa). VM-10 and VM-11 are samples from the second north flow (Fig. 1; 1N). Vegetation, including lichen and moss, covered a significant surface area of the samples, though it is highly variable, from no coverage at VM-03 to

entirely covered in 8 cm thick mat of moss at VM-08.

## 3.2  Selection and Preparation

VM-01-3, 06, and 08 through 11 (Table 1) were chosen for this study due to their higher modal abundance of mm-sized olivine grains (~5 %) compared to samples VM 04, 05, 12, and 13. All hand samples were scrubbed of vegetation and rinsed in water. Hand samples were jaw-crushed to mm sized grains. Matrix-free, mm-sized olivine was handpicked from the crushate based

on the distinct lack of cleavage and green color.

**Table 1** Sample Information

| Sample Name | Latitude | Longitude | Elevation (m) | Sample Thickness (cm) |
|---|---|---|---|---|
| VM-01 | 62.92311 | -137.37750 | 1052 | 3 |
| VM-02 | 62.92302 | -137.37790 | 1067 | 3 |
| VM-03 | 62.92302 | -137.37790 | 1067 | 3.5 |
| VM-06 | 62.90544 | -137.43741 | 718 | 3 |
| VM-08 | 62.90447 | -137.43590 | 725 | |
| VM-09 | 62.90413 | -137.43684 | 725 | 2 |
| VM-10 | 62.95478 | -137.36848 | 788 | |
| VM-11 | 62.95470 | -137.36888 | 785 | |



**Figure 1. Geologic Map of Volcano Mountain (Nelruna)** Crush-fusion analyses were done using powdered olivine xenocrysts from VM-02, 06, 08, 09, 10, and 11. Powdered olivine from VM-01 and VM-03 was step-heated. $^3He_c$ exposure ages are reported in Table 4.



## 4 Analytical Methods

### 4.1 Crush-Fusion


Handpicked olivines were crushed under vacuum for 2 min, and then re-crushed for 5 min following Blard et al. (2008). Prior to dropping samples into the crusher anvil, 2 min "no sample" crushes were measured and used to blank correct He measurements. Maximum blank crush levels of $^3$He and $^4$He were 1.88 x 10$^{-3}$ Matoms (0.07 fcc STP) and 5.38 x 10$^3$ Matoms (0.2 ncc STP) respectively (1 ncc = 2.6884 x 10$^4$ Matoms; STP are the standard temperature and pressure conditions of 273.15 K and 101.3 kPa where 1 mol of an ideal gas occupies a volume of 22419 cm$^3$). For some samples, recovered material from crushes was powdered in a mortar and pestle under ethanol and sieved to <30 μm, dried, weighed, and wrapped in Sn foil packets for fusion analysis. For other samples analyzed by fusion the hand-picked olivines did not undergo the initial vacuum-crush step, only powdering under ethanol. Ethanol was used to circumvent trapping of atmospheric He during powdering (Cox et al., 2022).


Sn-wrapped samples for fusion analysis were loaded into the dropper arm of a double wall furnace and evacuated overnight without baking. Samples were dropped into a carbon liner in the furnace and heated to ~1200 °C for 25 min. Re-extracts were measured directly after each extraction to ensure the complete release of helium. Two analyses were fused with the help of a lithium borate flux at ~1000 °C for 25 min in a molybdenum liner. The flux was used in these later fusions to reduce blank He levels by lowering extraction temperature (Farley et al., 2020). Hot blanks at ~1200 °C for the empty furnace were measured


for powder fusions. Maximum hot blank levels of $^3$He and $^4$He were 2.15 x 10$^{-3}$ Matoms (0.8 fcc STP) and 5.38 x 10$^3$ Matoms (0.2 ncc STP) respectively. Although blank levels are insignificant (<5 %) compared to levels of extracted helium, blank corrections were still made to all He measurements.

### 4.2 Step-heating

About 750 mg of powdered (<30 μm) sample was wrapped in Sn foil packets that were then loaded into the dropper arm of a


double walled furnace that was evacuated overnight without baking. The Sn packets were then dropped into the furnace, and He released at sequential low, middle, and high temperature steps. Temperature steps held for 30 min at 800 °C, 1000 °C, and 1400 °C were chosen for this experiment based on previous studies that observe the release of cosmogenic $^3$He below 1000 °C (Kurz, 1986a; Swindle et al., 2023). A thermocouple and pyrometer were used to calibrate furnace temperatures as a function of furnace output power.


Hot blanks were performed on an empty furnace at each of the scheduled temperature steps to use as blank corrections on He quantities at each step. Blank levels were <1.5 % of extracted He for all steps. Re-extracts were measured on the material after step-heating to ensure total helium release.

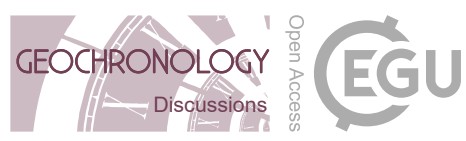

### 4.3 Helium Measurement

Helium was purified and measured as previously described (Horton et al., 2019; Swindle et al., 2023). Gas extracted from the furnace was passed through a charcoal U-trap to remove contaminants such as $CO_2$, $H_2O$, and Ar. Remaining reactive gases were removed using a hot SEAS NP10 getter and $H_2$ was removed using a cold SAES NP10 getter. Helium was then cryofocused on charcoal at 14 K and released into a Helix SFT mass spectrometer at 34 K. $^3$He was measured on a pulse-counting electron multiplier and $^4$He on a Faraday cup. Instrument sensitivity and internal consistency was monitored by

running a standard containing ~1.882 x $10^6$ Matoms (~70 ncc) of He ($^3$He/$^4$He ratio = 2.05 $R_A$) throughout our experiments. This standard was created from pure $^3$He and $^4$He using a capacitance manometer and the amounts delivered are known to better than ±1 %.

### 4.4 Shielding Correction

Corrections for snow cover, vegetation cover, and self-shielding were considered for every sample. The snow correction factor

was calculated based on Eq. 3 in Vermeersch (2007) based on the spallogenic neutron attenuation length, average snow thickness, and snow density. The sample area experiences on average 28.8 cm of snow cover for 5.5 months of the year (Government of Canada, 2007). A snow density of 0.3 g/cm$^3$ was assumed. Topographic shielding corrections were made for VM-01, 02, and 03 due to their location inside the VM cinder cone (see red outline in Fig. 1). Topographic shielding for the other samples was negligible.

## 5 Results

### 5.1 Crush

Results are shown in Table 2. Early measurements focused on samples with the most abundant separable olivine. Vacuum crushing of VM-06 and VM-09 revealed a mantle $^3$He/$^4$He ratio of 8.1 ± 0.2 and 8.3 ± 0.2 $R_A$ respectively. $^4$He concentrations were 2.03 x $10^6$ ± 2.34 x $10^4$ and 2.86 x $10^6$ ± 3.92 x $10^4$ Matoms/g (76 and 106 ± 1 ncc STP/g).

### 5.2 Fusion

Two crush and six powder fusion analyses of olivine xenocrysts from Volcano Mountain were measured in this study (Table 2). A large fraction of mantle He was obtained from all powder fusions, with $^4$He concentrations that range from 1.04 x $10^6$ ± 2.14 x $10^4$ and 2.93 x $10^6$ ± 6.04 x $10^4$ Matoms/g (39 ± 1 to 110 ± 2 ncc STP/g). $^3$He/$^4$He ratios of fusions ranged from 8.0 to 9.6 ± 0.3 $R_A$, with most fusions having slightly higher ratios compared to those obtained by crushing. Given that the crush

ratio of 8.1 ± 0.2 $R_A$ is likely representative of all VM samples, there are six crush-fusion pairs that can be used to calculate $^3$He$_c$ concentrations using Eq. 5. The highest concentration of $^3$He$_c$ we measured was 2.58 ± 0.65 Matoms/g (0.1 ± 0.02 pcc/g)





in VM-08 while the lowest concentration was within error of zero in VM-09 (Table 4). The mean concentration is $1.33 \pm 0.68$ Matoms/g ($0.05 \pm 0.02$ pcc/g).

**Table 2** Crush-fusion Results

| Analysis | Sample | Mass (g) | $^3$He (pcc/g) | ± | $^4$He (ncc/g) | ± | $^3$He (Matoms/g) | ± | $^4$He (Matoms/g) | ± | $^3$He/$^4$He ($R_A$) | ± |
|---|---|---|---|---|---|---|---|---|---|---|---|---|
| Crush | VM-06 | 0.63 | 0.85 | 0.02 | 76 | 1 | 22.8 | 0.5 | 2.03E+06 | 2.34E+04 | 8.1 | 0.2 |
| Crush | VM-09 | 0.10 | 1.22 | 0.02 | 106 | 1 | 32.8 | 0.7 | 2.86E+06 | 3.92E+04 | 8.3 | 0.2 |
| Fusion | VM-02 | 0.25 | 1.31 | 0.03 | 109 | 2 | 35.3 | 0.7 | 2.93E+06 | 6.04E+04 | 8.7 | 0.3 |
| Fusion | VM-06 | 0.50 | 0.51 | 0.01 | 39 | 1 | 13.8 | 0.3 | 1.04E+06 | 2.14E+04 | 9.6 | 0.3 |
| Fusion | VM-08 | 0.32 | 0.70 | 0.01 | 53 | 1 | 18.7 | 0.4 | 1.43E+06 | 2.93E+04 | 9.4 | 0.3 |
| Fusion | VM-09 | 0.07 | 0.51 | 0.01 | 46 | 1 | 13.7 | 0.3 | 1.24E+06 | 2.71E+04 | 8.0 | 0.2 |
| Fusion | VM-10 | 0.12 | 0.53 | 0.01 | 47 | 1 | 14.4 | 0.3 | 1.27E+06 | 2.62E+04 | 8.2 | 0.3 |
| Fusion | VM-11 | 0.10 | 1.15 | 0.02 | 100 | 2 | 30.9 | 0.6 | 2.70E+06 | 5.53E+04 | 8.3 | 0.2 |

This same fusion data can be cast as an isochron provided we assume the flows have the same exposure age and mantle component. Although field observations heavily suggest that these flows do not have the same exposure age, it is curious to test these assumptions. Regression of the $^3$He (pcc/g) and $^4$He (ncc/g) concentration data from our VM fusions reveals a

($^3$He/$^4$He)$_m$ ratio of $8.1 \pm 0.3$ $R_A$ (Fig. 2), which is consistent with our crush analyses of VM-06 and VM-09. The cosmogenic concentration estimated by the y-intercept of this isochron is $1.34 \pm 0.54$ Matoms/g ($0.05 \pm 0.02$ pcc/g) of $^3$He$_c$ (Fig. 2).

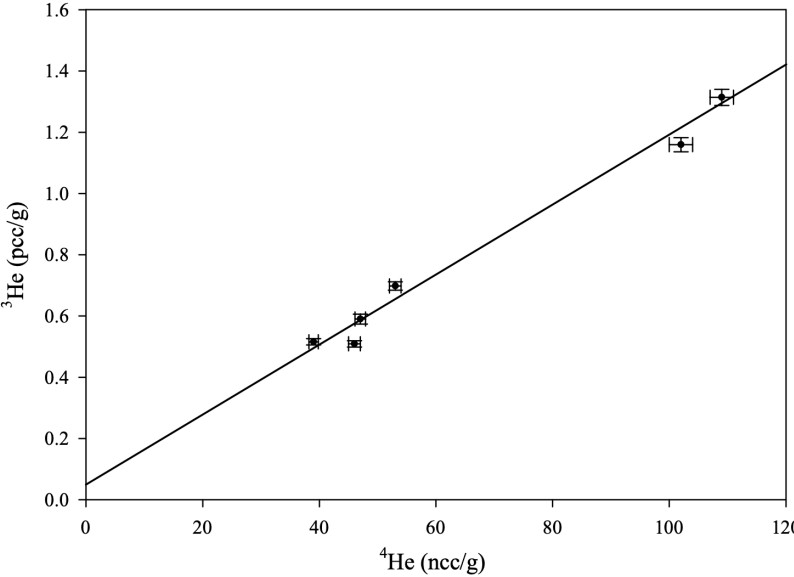

**Figure 2** Helium isochron (Blard, 2021) of powdered fusion data from VM-02, 06, 08, 09, 10, and 11. Assuming these flows have the same

exposure age and mantle component, the regression of $^3$He (pcc/g) vs $^4$He (ncc/g) yields a y-intercept that estimates the $^3$He$_c$ concentration



in pcc/g and a slope equal to ($^3$He/$^4$He)$_m$. This regression has a slope of 8.1 ± 0.3 R$_A$ and a y-intercept of 1.34 ± 0.54 Matoms/g of $^3$He. Regression was done using a MATLAB code created by Thirumalai et al. (2011) and is based on equations in York et al. (2004).

### 5.3 Step-heat

The low, middle, and high temperature steps of the VM-01 experiment yielded $^3$He/$^4$He ratios of 12.3 ± 0.2, 8.4 ± 0.2, and 7.9 ± 0.2 R$_A$ respectively (Table 3). A similar trend is observed in the VM-03 step-heat experiment where the $^3$He/$^4$He ratio evolved from 13.2 ± 0.4 to 9.0 ± 0.3, and then to 8.2 ± 0.2 R$_A$ in the low, middle, and high temperature steps respectively (Fig 3). The low temperature step in VM-01 released 92 % of the total $^3$He$_c$ and 80 % was released in the low temperature step in VM-03. Mantle $^3$He shows an opposite trend, with most mantle He released in the high temperature step in both step-heat experiments.

**Table 3** Step Heat Results

| Sample | Mass (mg) | Step T (°C) | $^3$He (pcc/g) | ± | $^4$He (ncc/g) | ± | $^3$He (Matoms/g) | ± | $^4$He (Matoms/g) | ± | $^3$He/$^4$He (R$_A$) | ± |
|---|---|---|---|---|---|---|---|---|---|---|---|---|
| | | 800 | 0.34 | 0.01 | 20 | 0.4 | 9.08 | 0.18 | 5.32E+05 | 1.07E+04 | 12.3 | 0.3 |
| VM-01 | 800 | 1000 | 0.29 | 0.01 | 25 | 0.5 | 7.86 | 0.16 | 6.78E+05 | 1.36E+04 | 8.4 | 0.2 |
| | | 1400 | 0.50 | 0.01 | 46 | 0.9 | 13.53 | 0.27 | 1.24E+06 | 2.49E+04 | 7.9 | 0.2 |
| | | 800 | 0.27 | 0.01 | 15 | 0.3 | 7.29 | 0.15 | 3.99E+05 | 8.03E+03 | 13.2 | 0.4 |
| VM-03 | 734 | 1000 | 0.25 | 0.01 | 20 | 0.4 | 6.75 | 0.14 | 5.39E+05 | 1.09E+04 | 9.0 | 0.3 |
| | | 1400 | 0.79 | 0.02 | 70 | 1.4 | 21.36 | 0.43 | 1.88E+06 | 3.78E+04 | 8.2 | 0.2 |


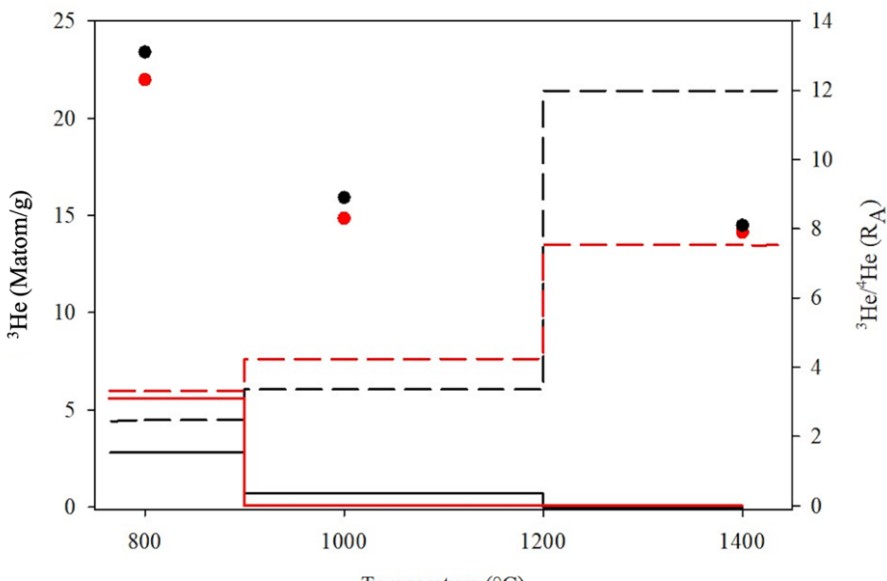

**Figure 3.** Evolution of $^3$He/$^4$He (circle), $^3$He$_c$ (solid line) and $^3$He$_m$ (dashed line) for VM-01 (red) and VM-03 (black) at the low, middle,



and high temperature steps. By the high-T step, all $^3He_c$ has been released. Most $^3He_m$ is released in the high-T step, highlighting how much mantle He is retained in VM olivine xenocrysts.

# 6 Discussion

## 6.1. Tightly Retained Mantle He in Volcano Mountain Olivine

Our crush-released $^3He/^4He$ ratios are within the typical range of mid-ocean ridge basalts (Graham et al., 1992), suggesting He in Volcano Mountain derives from the upper mantle. Regardless of its origin, the presence of this mantle He substantially complicates our cosmogenic $^3He$ measurement. While mantle $^4He$ concentrations of crushed olivines are rarely as high as the 2.9 x $10^6$ Matoms/g (~100 ncc STP /g) we obtained from VM-09 (Table 2) (Kurz et al., 1990; Farley and Neroda, 1998; Fenton and Niedermann, 2014; Heineke et al., 2016), the greater complication arises from the fact that mantle He is not effectively removed by powdering to <30 um. This is especially evident in samples VM-02 and VM-11, which have $^4He$ powder fusion concentrations as high as obtained by crushing (Table 2).

Powder fusions of VM-09, 10, and 11 have $^3He/^4He$ ratios within error of the crush ratios. The remaining fusions are marginally better, with $^3He/^4He$ ratios ranging from $8.7 \pm 0.3$ to $9.6 \pm 0.3$ $R_A$ (Table 2). The crush/fusion method relies on strong enrichment of the cosmogenic He component relative to the mantle component in the powder fusion. Our results suggest a mantle component that survives the powdering step, most likely <<30 μm fluid inclusions that remain intact until being melted during fusion. Evidence to support this are trails of voids (<10 μm) present in backscatter electron images of VM olivine xenocrysts (Fig. 4). These voids can be interpreted as secondary inclusion trails, which carry mantle He in them. Survival of the mantle component when crushing this fine is not typical, but it is very similar to results obtained on Twin Sisters (TS) peridotites (Swindle et al., 2023). Another possibility is that the ubiquitous kink banding and sub grain boundaries observed in TS and VM (Francis & Ludden, 1990) could contribute to mantle He retention.

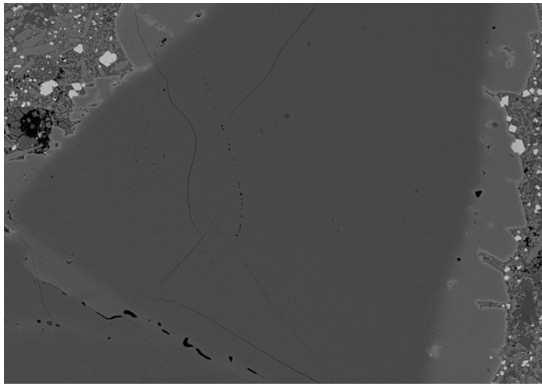

**Figure 4.** Electron backscatter image of olivine xenocryst in VM-08. The small voids interpreted as secondary inclusion trails are near the center of the grain.



## 6.2 High Degree of Separation in Low Temperature Step

The step-heat method is based on the release temperature of matrix-sited vs fluid inclusion-sited He. In low temperature steps, matrix-associated $^3$He is preferentially released while at the higher temperature steps, fluid-inclusions release mantle He. Three temperature steps allowed us to be flexible about the temperature that separates these two components. The evolution of the
$^3$He/$^4$He ratio with temperature thus represents a two-endmember mixing problem between mantle and cosmogenic He.

Compared to the highest $^3$He/$^4$He ratio from our dataset of powder fusions, the $^3$He/$^4$He ratios of the low temperature steps in our step-heat experiments are ~22-28 % higher. The percentage of cosmogenic 3He increases to 36-42 % when considering the lowest powder fusion $^3$He/$^4$He ratio. The degree of separation ($^3$He$_c$/$^3$He$_{total}$) can also be used to evaluate the resolution between cosmogenic and mantle He. The highest degree of separation in crush/fusion analyses was ~16 %. In contrast, the
degree of separation in the low temperature steps for VM-01 and VM-03 was 34 and 39 % respectively. It is possible that a more detailed step heat could yield even greater separation, but at the potential cost of analytical precision. In any case, the degree of separation obtained in these experiments provides a precise estimate of the exposure age of the VM flows.

## 6.3 Cross-Method Comparison of $^3$He$_c$ Concentrations

In addition to the crush-fusion analyses and He isochron, the step-heating protocol was applied to VM-01 and VM-03. Like in
the crush-fusion method, the measured $^4$He and the $^3$He /$^4$He$_{crush}$ ratio is used as a proxy for $^3$He$_m$ in each temperature step. The concentration of $^3$He$_c$ is calculated by subtracting the $^3$He$_m$ from $^3$He$_{total}$ for the low and middle temperature steps and then adding the cosmogenic components together:

$$^3He_c = (^3He_{low\ temp} - (^3He\ /^4He_{crush})(\ ^4He_{low\ temp})) + (^3He_{middle\ temp} - (^3He\ /^4He_{crush})(\ ^4He_{middle\ temp})) \quad (7)$$

The high temperature step was excluded because no $^3$He$_c$ was detected in this step. Equation 7 was used to calculate the
cosmogenic $^3$He concentration for the step-heat analyses, though it is worth noting that $^3$He$_c$ from the middle temperature step of VM-01 is negligible.

The $^3$He$_c$ concentrations calculated using the crush-fusion, isochron, and step heat methods are listed in Table 4 in both Matoms/g and pcc/g. Crush-fusion concentrations range from 0 to 2.58 ± 0.48 Matoms/g $^3$He$_c$ and the mean is 1.33 ± 0.65 Matoms/g for the 6 crush-fusion pairs. The He isochron method estimates a cosmogenic $^3$He component of 1.34 ± 0.48
Matoms/g and is within error of the crush-fusion mean. The step-heat method yielded the highest $^3$He$_c$ concentrations with the lowest uncertainties across methods (3.11 ± 0.28 and 3.50 ± 0.34 Matoms/g). Though VM-02 samples the same flow as VM-01 and VM-03, the uncertainty of the crush-fusion derived $^3$He$_c$ concentration is much greater (2.43 ± 1.26 Matoms/g).






**Table 4** $^3$He$_c$ Concentrations

| Method | Sample | $^3$He$_c$ (pcc/g) | ± | $^3$He$_c$ (Matom/g) | ± |
|---|---|---|---|---|---|
| Crush/Fusion | VM-02 | 0.09 | 0.05 | 2.43 | 1.26 |
| | VM-06 | 0.08 | 0.02 | 2.19 | 0.48 |
| | VM-08 | 0.10 | 0.02 | 2.58 | 0.48 |
| | VM-09 | 0.00 | 0.00 | 0 | 0 |
| | VM-10 | 0.004 | 0.02 | 0.11 | 0.56 |
| | VM-11 | 0.02 | 0.04 | 0.66 | 1.13 |
| Isochron | | 0.05 | 0.02 | 1.34 | 0.54 |
| Step Heat | VM-01 | 0.12 | 0.01 | 3.11 | 0.28 |
| | VM-03 | 0.13 | 0.01 | 3.5 | 0.34 |

## 6.4 $^3$He$_c$ Exposure Ages

Exposure ages were computed from $^3$He$_c$ concentrations and corresponding uncertainties (Table 5) using the CRONUS-Earth

online calculator (Balco et al., 2008; Phillips et al., 2016; version 3) where the production rate of $^3$He$_c$ is calibrated using data

from Borchers et al. (2016). Crush-fusion concentrations yield $^3$He$_c$ exposure ages ranging from 7.7 ± 4 ka to 11.1 ± 2.8 ka.

Using the $^3$He$_c$ concentration from the He isochron (Fig. 2) and assuming an average elevation of 800 m and a shielding

correction of 0.96, we calculate an exposure age of 5.2 ± 2.4 ka. In comparison, our step heat data yield $^3$He$_c$ exposure ages of

10.8 ± 1.3 ka and 11.1 ± 1.1 ka for VM-01 and VM-03 respectively. These ages are consistent with each other and have lower

uncertainty than the exposure ages calculated using the crush-fusion or isochron $^3$He$_c$ concentrations. Since VM-01 and VM-

03 are sampled from the same flow, we can average the ages to get 10.9 ± 1.1 ka.

Volcano Mountain comprises especially young flows and the material is unburied and uneroded. VM-01 and VM-03 are

samples from the youngest flow in the Volcano Mountain sequence. In this case, $^3$He$_c$ exposure ages can also interpreted as

formation ages for the flows, meaning the olivine in the youngest flow began retaining $^3$He$_c$ at Earth's surface 10.9 ± 1.1 ka.

This exposure age is older than the age estimated from lake sediments (7300 BP and 4200 BP) (Jackson and Stevens, 1992)

by a few thousand years. Interestingly, this age is coincident with end of the last glacial maximum of the Cordilleran ice sheet

(Menounos et al., 2009; Clague, 1989).

Based on the stratigraphic position of the lava flows it was expected that VM-01 and VM-03 would return the youngest

age. This method therefore confirms that the crush-fusion and isochron method are returning ages that are too young. It is

possible the isochron method underestimated the cosmogenic $^3$He component because the samples are different ages or because



they experienced different shielding. Results from the step heat method are considered the only reliable exposure ages derived from these experiments.

**Table 5** $^3$He$_c$ Exposure Ages

| Analysis | Sample | Elevation (m) | $^3$He$_c$ (Matoms/g) | ± | Shielding Correction | Age (ka) | σ |
|---|---|---|---|---|---|---|---|
| Step Heat | VM-01 | 1052 | 3.11 | 0.28 | 0.96 | 10.8 | 1.3 |
| Step Heat | VM-03 | 1067 | 3.50 | 0.34 | 0.96 | 11.1 | 1.1 |
| Crush/Fusion | VM-02 | 1067 | 2.43 | 1.26 | 0.96 | 7.7 | 4.0 |
| Crush/Fusion | VM-06 | 718 | 2.19 | 0.48 | 0.97 | 9.5 | 2.1 |
| Crush/Fusion | VM-08 | 725 | 2.58 | 0.65 | 0.96 | 11.1 | 2.8 |
| Crush/Fusion | VM-09 | 725 | 0.00 | 0.00 | | | |
| Crush/Fusion | VM-10 | 788 | 0.11 | 0.56 | | | |
| Crush/Fusion | VM-11 | 785 | 0.66 | 1.13 | | | |

## 285 7 Conclusion

Volcano Mountain nephelinites contain olivine likely sourced from disaggregated peridotite xenoliths. This olivine contains high concentrations of upper mantle He and powdering does not effectively remove this component. Swindle et al. (2023) were similarly unable to remove mantle He components by powdering olivine from the Twin Sister peridotite. In both cases, this mantle He may be housed in fluid inclusions too small to be released by crushing. Alternatively, ubiquitous kink bands and

subgrain boundaries seen in the Twin Sisters and Volcano Mountain flows could be the source of the He that survives powdering. In either case, survival of mantle He in VM powder fusions caused the crush-fusion and isochron methods to yield imprecise $^3$He$_c$ concentrations. By using the step heating method developed here, we were able to better isolate $^3$He$_c$ from $^3$He$_m$, allowing us to obtain an exposure age for VM of $10.9 \pm 1.1$ ka.

### Data Availability

All data is provided in the figures and tables of this manuscript.

### Author Contribution

J. M. Mueller prepared the manuscript with significant contributions from K. A. Farley and J. D. Bond. J. D. Bond and B.C Ward completed the field sampling.

### Competing Interests



The authors declare that they have no conflict of interest.

**Acknowledgements**

We respectfully acknowledge and thank the Selkirk First Nation (SFN) for their support and guidance on this project. Roger Alfred and Elli Marcotte (SFN) provided traditional knowledge of Nelruna while in the field, which enlightened and guided our efforts. Sampling assistance was provided by Leyla Weston (YGS), Keyshawn Sawyer (SFN) and Sofia Bond (YGS).

Funding for the field work was provided by the Yukon Geological Survey. Safe access to the area was provided by Malcolm Turnbull from Capital Helicopters. We would like to thank Dale and Sue Bradley from Pelly Ranch for their hospitality while completing the field work. This paper is assigned Yukon Geological Survey contribution number 066.

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
