# Peer review of "Cosmogenic 3He dating of olivine with tightly retained mantle 3He, Volcano Mountain, Yukon"

_Geochronology, 2024_

## Referee Comment (RC2)

Review of Mueller et al., (submitted to Geochronology)

Disclaimer: I have only access to the original manuscript (I cannot see the updated version from the previous review). However, it should not be an issue regarding my major comments section but I am sorry in advance if any redundancy for the detailed comments section.

This manuscript is a scientific contribution describing the $^3$He cosmogenic ($^3$He$_c$) extraction and determination from $^3$He mantle-bearing olivine. The paper shows that $^3$He$_c$ can still be extracted from highly enriched $^3$He mantle olivine using the step heating extraction method (Kurz 1986a). The authors are using olivine crystals sampled from 6 different lava flows from the Nelruna volcano (i.e. Volcano Mountain - Yukon). The authors analyzed the $^3$He total signal with two different extraction methods (crush/fusion, and step heating) in order to separate the cosmogenic $^3$He signal from the $^3$He mantellic component. Their results show that only step heating method is able to separate precisely $^3$He signal between the mantle and the cosmogenic components in these samples. Overall, I find the paper well written and easy to follow but some sections are lacking of detailed information (see major and detailed comments below). Moreover, after careful review of the paper, I have several concerns regarding the data interpretation, especially the crush/fusion and isochron dataset. I think there is some important consideration that the authors might want to look at, in order to get more detailed information on the mantellic source of the samples and/or the $^3$He$_c$ signal. I think more information could be retrieved from the crush/fusion data and could potentially reveal some signature from those tiny void inclusion, that deserve to be discussed (see major comments section below). In addition, considering the whole dataset (crush/fusion, isochron and step heating), I would consider that variable mantellic source signatures cannot be properly ruled out despite the fact that step-heating data seem convincing, and that $^3$He$_c$ component is potentially retrieved from those $^3$He mantle rich bearing olivine. However, I do have some concern regarding the low temperature (800ºC) step heating signal. This temperature step is crucial for the authors to successfully demonstrate that they extracted the $^3$He$_c$ signal. Therefore, considering the importance of this He data (at T~ 800ºC), one could think that the $^3$He/$^4$He signal (Ra ~12-13) could be potentially related to a more primordial mantle source preserved from their observed inclusion trails (< 10 μm) instead of the $^3$He$_c$ component. Indeed, the authors pointed out that a mantle component could be preserved in tiny (<10 um) void inclusion observed within the olivine crystal (see the electron backscatter image in Fig.4). If these tiny (<10 um) inclusions are preserving a deeper mantle source signature, the first temperature step would therefore preferentially extract signal from these inclusions (due to their very small dimension). Consequently, first temperature step signal would reflect the signature of the inclusions rather than the cosmogenic $^3$He. I think it is important to fully clear out that concern as this temperature step is critical for the paper and therefore should be fully investigated. Please find my detailed comments on that issue in the major comments section below.

The figures and tables show clear information but some sections are not well enough documented. Below I have listed my comments, with major issues within major comments section while secondary comments are located in the detailed comments section. To summary, the paper is well construct but require additional more careful work on the data treatment. A more detailed discussion regarding the first temperature step from the step-heating experiment is needed and additional information can be retrieved from the crush/fusion and isochron dataset. Nevertheless, the paper is providing interesting information and I do recommend this manuscript for publication only after major re-work and attention to their dataset.

**Major comments:**

**Crush/fusion experiment**. The authors are providing two data points for the crushing experiment (sample VM-06 and VM-09) with a Ra at ~8.2, and 6 data points for fusion with an average Ra at ~8.7. However, for 3 fusion samples (VM-02, VM-06 and VM-08) the $^3$He/$^4$He ratio (Ra) extracted is measured at 9.2 while the 3 other fusion samples (VM-09, VM-10 and VM-11) the $^3$He/$^4$He ratio (Ra) extracted is measured at 8.2, similar to the crush value. The authors also re-analyzed only two crushed samples (VM-06 and VM-09) with a one-step outgassing (fusion) and the results show either a crush-like Ra at ~8 (VM-09) or a higher Ra at ~9.6 (VM-06). Consequently, proper crush/fusion methodology can only be conducted on sample VM-06 and VM-09 as they are the only two samples with pre-crushed data. However, the authors seem to assume that crush Ra value from samples VM-06 and VM-09 can be applying to the 6 different lava flows dataset (line 165). I would mention that only two crush values used to assume the mantellic source of 6 different lava flow is somehow under representative to my opinion. I would have preferred a proper crush/fusion dataset for all considered olivine presented in this study in order to reduce any potential variability of the mantellic source. If considering that $^3$He extracted from fusion reflect the matrix component and thus release the cosmogenic $^3$He (with the assumption that $^3$He radiogenic is negligible), I would therefore expect that sample VM-06 (Ra ~9.6) either preserved more efficiently its $^3$He$_c$ or has an older age to build up a $^3$He cosmogenic signal. However, the authors sample description mentions that sample VM-09 (Ra ~8) is taken from the oldest lava flow analyzed in this study, and therefore I would have expected higher Ra value from sample VM-09 than VM-06 if $^3$He$_c$ signal was released preferentially during fusion. Additionally, surface vegetation coverage could hinder the production of $^3$He cosmogenic. However, the authors mentioned that sample VM-08, taken on the same lava flow as VM-09, was the most covered sample with an 8cm thick moss (line 100) but still gives a Ra value of 9.4, significantly higher than Ra signal on VM-09 (Ra ~8). Consequently, both the sample location and the vegetation cover don't seem to be a good explanation for the observed Ra values measured in

those samples. If we assume that the two crushed Ra values from sample VM-06 and VM-09 can be extended to the mantle component signature for all the samples, the discrepancy observed between samples VM-09, VM-10 and VM-11 (Ra ~ 8.2) and samples VM-02, VM-06 and VM-08 (Ra ~ 9.2) remains difficult to explain. The authors seem to not discuss those Ra values (lines 211-212) but I think there is some important information that deserved some attention. Moreover, this might have some implication for the step heating experiment results.

A possible alternative hypothesis to explain the observed dataset could be that the olivine samples are preserving a less degassed (i.e. deeper) reservoir signature than MORB (Ra > 8) from tiny inclusions within the olivine crystal. Those tiny inclusions (< 10 μm), observed by the authors (lines 215-219) could bear an OIB-like mantle signature (Ra > 8) or a more variable mantellic composition (larger Ra range), and therefore the variability between measured samples could be explained with a more or less amount of those tiny inclusions within the olivine matrix. Consequently, this could reflect a different mantellic source composition between the different samples rather than $^3He_c$ component. This would better explain the Ra value variation in the fusion samples. On a first order check, we can verify if the isochron conditions are verified following the simple case of $^4He$ purely magmatic (no radiogenic $^4He$, as assumed in the paper). The isochron equation is given Blard and Pik (Chemi. Geol. 2008) as: $(^3He/^4He)_{fusion}$ = $^3He_c$ x $1/^4He_{fusion}$ + $(^3He/^4He)_{magmatic}$. If the validity of the equation is verified, therefore the isochron should display a linear regression. I quickly plotted below the data from the paper using the above equation (Fig. 1):

[Figure]

Fig. 1

The plot above does not show any convincing linear regression to my opinion, meaning that the assumptions are not met for the validity of the isochron. This could reflect either a variable $(^3He/^4He)_{magmatic}$ ratios in the different samples, an inhomogeneous $^3He_c$ concentration or a similar $^4He$ magmatic concentration among the olivine crystals. If the samples have a variable mantle component (i.e. variable $(^3He/^4He)_{magmatic}$ ratios) therefore the isochron plot can be interpreted as a mix between MORB like mantle and a deeper un-degassed signal (OIB-like) contained in the tiny (<10 μm) void inclusion. That could lead to the observed $^3He/^4He$ signal in their fusion samples as the concentration between the different samples are dependent of the crystal size (and not controlled by $^3He_c$). Indeed, if we observe the different masses of the olivine crystals, we can see that larger (or bigger) olivine crystal systematically lead to higher Ra value. For example, sample VM-02 (m = 0.25g), VM-06 (m = 0.5g) and VM-08 (0.32g) have an average Ra at ~9.2, while sample VM09 (m = 0.07g), VM-10 (m = 0.12g) and VM-11 (m = 0.1g) have an average Ra at ~8.2, in agreement with the assumption of a size/mass dependency of the Ra value due to the contribution of tiny inclusions (if we assume that the tiny inclusion concentration is more or less similar between samples). In addition, radiogenic component $^4He*$ is not mentioned in the paper but could also affect the $^4He$ signal measured from the olivine, as the last temperature step from the step heating experiment on sample VM-01 shows a Ra value of 7.9, lower than the value given by the crush dataset of Ra ~8.2. This potential $^4He*$ contribution could impact the step-heating temperature as Kurz 1986a showed that a very small fraction (<1%) of $^4He*$ on the first heating step can lead to a lower Ra signal. Consequently, the determination of the $^3He_c$ signal could be likely underestimated due to the low resolution of the step heating experiments (3 temperature steps with only one at low T). This could lead to underestimate ages for the associated lava flows. This is not discussed by the authors while it can be important for the $^3He/^4He$ ratio.

Overall, the crush/fusion methods can only be properly applied on two samples, and those two samples (VM-06 and VM-09) show Ra values that are difficult to explain with a purely $^3He_c$ signal at the moment. In addition, considering all the data with the assumption that $(^3He/^4He)_{crush}$ ratio from samples VM-06 and VM-09 is representative for all the samples, the Ra values are still difficult to explain with a purely $^3He$ cosmogenic signal. Therefore, a variable $(^3He/^4He)_{magmatic}$ ratio could better explain the variability of the Ra values. This variability in the mantellic signature could originate from a deeper mantellic source within the tiny inclusion (<10 μm) that are

not extracted by the crush experiments due to their small size but are preferentially extracted from the temperature steps due to their very small diffusion domain. Therefore, the Ra variability observed in the fusion samples could simply reflect the variability of the tiny inclusion concentration between the different olivine. This agrees with the observed mass and Ra values where higher Ra are systematically observed with higher masses (i.e. higher numbers of tiny inclusions). As a consequence, the first temperature step on the step-heating experiments could therefore not reflect the $^3He_c$ signal, but rather reveal this OIB-like mantellic component from the tiny inclusions.

**Step heating experiment.** The major findings of the paper are provided by the 3 temperature step heating experiments ($T_1 = 800$ºC, $T_2 = 1000$ºC and $T_3 = 1400$ºC), and especially the first temperature steps ($T_1 = 800$ºC). The authors analyzed samples VM-01 and VM-08 with this protocol and revealed a $^3He/^4He$ (Ra) at ~12.3 and 13.2 respectively. This ratio is significantly higher than the one recorded by the crushing (Ra ~8.2) or the fusion (Ra ~ 8.7) experiments on average. The author interpretation is that higher Ra value reflect $^3He$ cosmogenic enrichment and therefore this Ra is the precise determination of the $^3He_c$ component is those $^3He$ mantle-bearing olivine. The two other temperature steps provide Ra value near ~8 and contribute little ($T_2$) or none ($T_3$) to the $^3He$ cosmogenic signal. The authors interpretation is that Ra value around ~8 reflect a MORB signature associated to a mantle component while higher Ra values are induced by $^3He$ cosmogenic production. This is supported by the crushed samples (VM-06 and VM-09) that give a Ra value at ~8.2 and reflecting the trapped magmatic $^3He$ (i.e. fluid inclusion). First observation is that I wonder why they didn't apply this methodology to all of their samples as it seems the only way to determine $^3He_c$ from these high He mantellic-bearing olivine. I would have expected, following their result on crush/fusion experiment, that they would have also re-analyzed the other samples to properly determine the $^3He_c$ and thus their associated ages, especially knowing that the authors took several samples for redundancy purpose (line 87). Secondly, as mentioned in the above sections, a variable mantellic source cannot be ruled out to explain the observed data from the crush/fusion and isochron dataset. Therefore, a variable mantellic composition that affecting the $^3He/^4He$ signal can also be explained with the step-heating experiment as this deeper mantellic component is assumed to be preserved into small trails inclusion (<10 μm). We can therefore assume that this signal is revealed preferentially at low temperature due to the inclusions very small diffusion domain (<10 μm). A way to test this hypothesis would be to perform a more detailed step heating experiment, at low temperature, to investigate if signal from a lower temperature (T ~500ºC) deplete those tiny inclusions and allow $^3He$ cosmogenic signal to be released at slightly higher temperature (T~800ºC). Such detailed temperature step protocol for $^3He_c$ signal on olivine powder has been described in Kurz 1986a to separate $^3He/^4He$ signal from very small faction of $^4He$ radiogenic signal implanted on the phenocryst surface (<1%). The $^3He/^4He$ ratio on lower temperature (T~500ºC) tend to have a lower Ra than the successive step (T~800ºC), as the first temperature step is likely more affected by $^4He$ radiogenic component (see Kurz et al., 1986a). In this manuscript, the signal at T~800ºC could also be affected by such radiogenic component or/and potentially by the tiny high-Ra mantellic inclusions. In both case, lower temperature step should either display lower Ra value if $^4He^*$ is implemented on the surface (even a very low fraction <1%) or higher Ra if reflecting preferentially the tiny inclusion contribution. The second temperature step (at T~800ºC) should therefore reflect a better $^3He_c$ signal if any. However, as this study present only one low temperature step ($T_1$), I would recommend to be careful with the signal at T~800ºC as additional processes affecting the Ra value cannot be ruled out. Consequently, lava flow ages presented in this study can be either under or over estimated. Additional important information can also be linked to processes in the mantle source (if we assume a variable mantellic component) of the analyzed lava flows (with a component from deeper mantellic source affecting the volcanic region). A more global geological implication could result from the paper and provide more insight of the volcanic system below the Nelruna volcano and its volcanic province.

**Isochron method.** In addition to the crush/fusion dataset, the authors are using the isochron method to extract a $^3He_c$ value from the fusion dataset. At line 172, they specify: "we assume the flow to have the same exposure age" while they specify between lines 93 and 98 that the lava flows have stratigraphy different ages from the youngest flow (near the cinder cone) to the oldest flow (third south flow). Therefore, samples collected on the different lava flows are expected to have different exposure ages. The authors are aware of such issue, (lines 173-174), but still try to apply the isochron method. I think a better approach would have been to use different aliquots of the same olivine population (sampled from the same lava flow) and cast these data into a proper isochron as defined by Blard and Pik (Chemi. Geol. 2008). That could have provided more reliable information on the potential $^3He_c$ signal. Therefore, no isochron is technically showed in this paper and I would be caution on the term used here. Instead, the authors are using 6 different lava flows merged altogether. The initial assumption of identical $^3He_c$ concentration is then not valid and a linear regression is not expected from the isochron definition given by Blard and Pik (Chemi. Geol. 2008). On the other hand, the authors are providing a plot with $^3He$ (pcc/g) vs. $^4He$ (ncc/g) following the relationship: $^3He_{fusion} = {}^3He_c + {}^4He_{fusion}*(^3He/^4He)_{crush}$ (*equation 5 in the paper*). First, I want to point out that the data in table 2 display some discrepancies and are not matching the one plotted in Fig. 2. The exact Ra calculated from the $^3He$ and $^4He$ data are slightly different. Ra values for samples VM-06, VM-08, and VM-10 are given at 9.6, 9.4 and 8.2, respectively in table 2 while calculated values for these samples (directly from $^3He$ and $^4He$ data in table 2) yield to Ra values of 9.4, 9.5 and 8.1, respectively. Therefore, when plotting $^3He$ and $^4He$ directly with the data from table 2, I obtain a slightly different plot of the Fig. 2. See below the original Fig. 2 (left plot) and the revised Fig. 2, noted here Fig. 2a (right plot) with its associated linear regression equation and an estimated error envelop (from a regular linear regression fit in Excel):

[Figure]

[Figure]

Fig. 2

Fig. 2a

While I am not sure why some plotted data in Fig. 2 are not the one provided in table 2, I am assuming that $^3$He and $^4$He dataset in table 2 are the most reliable data source and I will consider then Fig. 2a as the best data representation. This slight discrepancy has, however, an important implication. Therefore, if we assume that this linear regression follows the equation 5 for Fig. 2a, the regression line at x = 0 should provide the $^3$He cosmogenic value, calculated at 0.0266 pcc/g or 0.72 Matoms/g (see equation on the Fig. 2a). Note that this value is significantly lower than the one estimated by the authors (1.33 Matoms/g). I am not sure why I obtain such different value from the authors (despite the data discrepancy noted above). Please check if that estimation is correct. I also estimated the errors by taking an upper and lower envelop of the regression line to fit at best the dataset (see Fig. 2a). Then, the estimated error is 0.72 $^+$/- 2 Matoms/g. This is a very large error for such an apparent good linear fit and much larger than the one given by the authors (1.33 $^+$/- 0.68 Matoms/g). Therefore, some additional processes that affecting this dataset is likely to happen. An alternative approach is to consider that olivine crystals are actually representing two separate group sampling two different mantellic components (a MORB-like at Ra = 8 and an enriched Ra source). Therefore, samples VM-02, VM-06 and VM-08 (Ra ~9.2) representing the higher Ra source while samples VM-09, VM-10 and VM-11 (Ra ~8.2) representing a MORB-like source (see Fig. 2b below).

[Figure]

Fig. 2b

Data from the two crushed samples are also represented (solid diamond) but are not included in the regression calculations. If we also assume that no $^3$He cosmogenic data is present in those sample, therefore the Y intersect is set to 0. The two linear regressions from this plot describe significantly better the observed data and allow to separate the two different mantle components, a MORB-like component (solid line) at ~8.2 (slope = 0.0114) for samples VM-09, VM-10 and VM-11 (solid circle) in agreement with the crush values (Ra ~8.2, solid diamond datapoint), and an OIB-like component (dashed line) at 8.9 Ra (slope = 0.0123) for samples VM-02, VM-06 and VM-08 (solid square) in agreement with Ra measured in those samples (Ra ~9.2) within errors. This agrees well with the hypothesis of a variable mantellic sources developed earlier (see major comments above). However, one could also consider that $^3$He$_c$ signal is responsible for the enriched Ra value observed in sample VM-02, VM-06 and VM-08. In such case, if we consider some $^3$He$_c$ in those samples, therefore the condition where the Y intersect is set to 0 is no valid anymore and the regression line for samples VM-02, VM-06 and VM-08 is shown in Fig. 2c below:

[Figure]

Fig. 2c

Consequently, important information could be retrieved from this Fig. 2c to support a $^3He_c$ component. First, linear regressions exhibit similar slopes giving a Ra at 8.16 for samples VM-02, VM-06 and VM-08 and Ra at 8.24 for samples VM-09, VM-10 and VM-11. However, while the slope value agrees with the crush value (Ra ~8.2), samples VM-09, VM-10 and VM-11 show higher Ra values (up to 9.5 Ra) which could then reveal the cosmogenic component. Therefore, the regression line at x = 0 could reflect an average $^3He_c$ value calculated at 0.0845 pcc/g or ~2.3 Matoms/g while regression line from sample VM-09, VM-10 and VM-11 gives no $^3He_c$ signal. This value could be a good estimate for average $^3He_c$ component for samples VM-02, VM-06 and VM-08. Secondly, $^3He_c$ value calculated with equation 5 (line 54) from the crush/fusion method for samples VM-06 give a value at ~2 Matoms/g. VM-02 and VM-08 do not have crush data but if we assume an average crush value, then $^3He_c$ is estimated at ~2.3 and ~2.8 Matoms/g, respectively, in fairly good agreement with VM-06 and the regression line values. Although this interpretation of the data cannot be ruled out, it has to be explained in the light of the sample location (i.e. expected lava flows exposure age), which still do not have a satisfactory explanation. The hypothesis of a variable mantellic source remains more likely to explain the data presented here, according to me. Consequently, the dataset provided by the authors is quite unclear to properly determine if $^3He$ cosmogenic is observed rather than a mix between two mantle sources. I would recommend to do a proper isochron (i.e. samples from same olive population), and calculate the pseudo-isochron with R-value determination (see Blard and Pik, 2008) to include the $^4He$ variability from different magmatic $^4He$ concentration (as well as radiogenic $^4He^*$), to better investigate the $^3He_c$ signal from these samples. Such simple $^3He$ vs. $^4He$ plot is not enough to capture the complexity of processes that can influence the He content in those olivine crystals.

**Detailed comments:**

The article seems to point out that they developed a new methodology to analyze $^3He$ cosmogenic signal by step heating the samples (see lines 68-69, and 292). This method, however, is not recent and has been used and developed by other authors before (Kurz 1986a). It would be more appropriate to refer at the step heating method as an already known approach (such as the crush/fusion or the isochron) but mentioning that you have adapted it with three temperature steps (800, 1000 and 1400ºC).

**Line 20:** "precise estimates of cosmogenic" is not really supported by the original manuscript as the step heating experiment (where the $^3He_c$ is the best measured) shows only one low temperature step, which already limit the precision of the $^3He_c$ determination. Indeed, very low (<1%) $^4He^*$ contribution (Kurz 1986a) or a variable mantle component cannot be investigating with such low-resolution step heating, and could likely affect the $^3He/^4He$ ratio.

**Line 66:** I disagree here. The isochron method was not used in the paper, otherwise aliquot from same olivine population would have been used. In addition, the equation for the isochron given by Blard and Pik (2008) is not used here. The authors simply plot $^4He$ vs. $^3He$ following equation 5 in the text which correspond to the classic crush/fusion method and called that an isochron. I would either remove isochron method statement, or if you want to assume that the 6 lava flows have same age, and can be used as one population, it should be clearly stated and the proper isochron equation should be used to verify the validity of the method. Please see my major comment above for more detailed discussion on the crush/fusion and isochron dataset.

**Line 68**: "we developed" should be replaced by "we used" or similar phrase.

**Line 91-92**: The authors specified that they have additional notes on the samples, such as vegetation cover or average sample depth, but failed to provide those data (or I didn't see them). I would have like to get them in a supplementary material, specially that some detailed sample information could have be beneficial.

**Line 102-103:** The authors specify in this section that 4 samples are disregarded and 8 samples are selected due to the mm-sized olivine. Could you please specify if no mm-size olivine were found in those 4 lava samples or if the olivine quality was insufficient for proper $^3He_c$ investigation? It is unclear from the description why those samples are disregarded. In addition, where is sample VM-07? It is supposed to be used in the study (see line 96 and line 103) but I cannot find this data in the text, the tables or the plots.

**Table 1**: This table could be more interesting if more information regarding the sample notes were included such as the vegetation cover or the topography shielding.

**Line 111**: The authors specify that the crushing protocol is derivated from Blard et al., 2008. There is no Blard et al., 2008 in the reference list. I found however, Blard and Pik (Chemi. Geol. 2008) and Blard and Farley (EPSL 2008). Please be careful when referencing literature (see comment at line 182 as well). In any case, both Blard and Pik (Chemi. Geol. 2008) or Blard and Farley (EPSL 2008) papers do not contain any indication (or I couldn't find any) for a crushing step at 2min followed by a step at 5min for proper $^3He_c$ extraction. Could you please provide information where this protocol has been taken? In addition, early crushing steps can potentially release $^3He_c$ from the matrix (see Blard and Pik (Chemi. Geol. 2008) and Blard and Farley (EPSL 2008) papers for more details). Do you have investigated such $^3He_c$ loss/contamination on the $^3He/^4He$ crush signal? This could lead to important impact on $^3He_c$ measurements if not well estimated. It should be at least mentioned here.

**Line 121-122**: Please add the re-extraction data to the dataset (in the main text or in supplementary). I noticed also that fusion is performed at ~1200ºC for 25min but step heating experiments show un-degassed samples at T~1400ºC for 30min step (last temperature step at table 3). I would recommend therefore to provide all the re-extract dataset to ensure that the fusion and step heating samples have been properly outgassed. At the moment, it seems difficult to fully outgas the samples with one temperature step at ~1200C for 25min. If re-extraction shows significant He, did you then add them to the total? Please provide additional information.

**Line 124-125**: The authors are using a blank correction for the He analysis with an empty furnace while using Sn foils to wrap their samples. I would have expected blanks to be run with empty Sn-foil packet instead, to better account the blank value. Same comment can be given for step heating experiment (see line 135). Given the blank level is given at 5% (without Sn-foil contribution), I suspect that blank could be underestimated if He outgassing for the Sn foil is not accounted for. In addition, 5% He blank contribution is not insignificant to my opinion.

**Line 134-135**: Please provide the re-extract data.

**Line 157**: What means early measurement here?

**Line 166**: Using equation 5, for sample VM-08, I calculate a $^3He_c$ at 2.8 Matoms/g instead of 2.58 Matoms provided by the authors. I am using an average $^3He/^4He_{crush}$ at 1.12x10$^{-5}$ (from VM-06 and VM-09). I suspect that you are either using VM-06 or VM-09 crush value but without any justification. Therefore, for the sake of consistency, I would recommend to use VM-06 and VM-09 average crush value for all VM samples at the exception of VM-06 and VM-09 where $^3He_c$ can be properly determined. Otherwise please justify which crush values you are using for all VM analyzed samples.

**Table 2**: For clarity purpose, I would add a special label for the two samples that have coupled crush/fusion data. The other could be labelled as uncrushed olivine.

**Fig. 2**: Please give the linear regression value in the plot. This is important information and should be display clearly in the plot, not in the caption. Also, as mentioned in my major comments above, the data plotted here is not representing the data in table 2. Please check why this is not the same as this might lead to different $^3He_c$ given by this regression line based from equation 5.

**Line 182:** Thirumalai et al., (2011) and York et al., (2004) are not listed in the reference section. Please be sure all the references are included.

**Fig. 3**: The red solid line for $^3He_c$ is likely to be wrong. The calculated $^3He_c$ from equation 5 using VM-1 data at T~800ºC lead to $^3He_c$ calculated at 3.1 Matoms/g (which represent 92% of the total $^3He_c$ for VM-01), but in the figure, the red solid line is showing a value >5 Matoms/g. The black solid line seems ok, but could you please check if the plot has the proper values calculated for $^3He_c$? In addition, the last temperature step (T~1400ºC) is very likely not fully outgassed. The signal is still showing high He content. If you have the re-extraction, please provide them to ensure that total outgassing of the sample is performed.

**Lines 240-241**: I think the argument for more detailed step heating is important here. Knowing that early temperature step can contain important information, especially for $^4He^*$ contribution that can lead to a lower Ra

value at low T-step (see Kurz 186a), or if we suspect some $^3$He/$^4$He lower mantle contribution from the tiny inclusions. I would have therefore, expected a better resolution for the step heating experiments. The SFT is capable of analyzing significantly lower values of $^3$He, much lower than ~0.4 pcc/g ($^3$He concentration given at T~800°C), especially knowing that blanks value is given at ~0.8 fcc, and therefore, in the worst-case scenario, acceptable values for $^3$He could be potentially measured as low as 0.008 pcc (~10 times the blank). However, as a theoretical example, if a lower temperature step (T~500°C) is performed on sample VM-01, and if $^3$He signal is measured 100 times lower than the one at T~800°C (i.e. ~0.07 Matoms), therefore the blank contribution given at 0.002 Matoms should "only" represent ~3% of the signal, which is quite acceptable. $^4$He signal, on the other hand is more complicated and could lead to some limitation under the current analytical blank of the double wall furnace. The $^4$He blank is given at 0.2 ncc which limit the measured signal at ~2 ncc of $^4$He (~10 times the blank to ensure sufficient precision). It is noteworthy that signal lower than 10 times the blank can be measure but then larger error is expected and could limit the interpretation. Nevertheless, $^4$He signal 10 times lower than the measured ones at T~800°C (i.e. $4.3 \times 10^4$ Matoms) could still be measured for a hypothetical step at T~500°C. Blank error could account for ~12% (5380 Matoms). I would like, however, to point out that the double-wall furnace (where step heating experiment is performed) is not baked but solely pumped out overnight (see lines 120-121), and $^4$He blank given by the authors are quite high compared to some double wall furnace blanks given by other studies (Blard et al., 2015; Kurz, 1986; Williams et al., 2005, Yokochi et al., 2005, Zimmermann et al., 2012; 2018, Zimmermann and Marty, 2014), where blank values are given as low as ~300-600 Matoms. If we assume that blank values can be reduced significantly in the range of 0.02 ncc (~600 Matoms), then previous blank contribution of ~12% (for a hypothetical T~500°C step) will drop at ~1.5%, and $^4$He concentration 100 times lower could be even analyzed. In addition, peak jumping analyses can also be performed to measure $^4$He on the CDD to keep good $^4$He measurement precision for very low signals. Consequently, I think the authors could safely performed a more detailed step heating experiment with limited loss of the analytical precision given by the SFT capacity and/or a better baking/cleaning/analytical protocol for $^4$He analysis.

**Line 245-247**: Why the authors are using the crush value here, while they have the mantle component determined with the highest temperature (T~1400°C, where no cosmogenic contribution is estimated, see line 249)? I would rather use the high temperature $^3$He/$^4$He ratio for the mantle composition instead of using a crush value from another lava flow (VM-06, VM-09 or average). My understanding is that no crush step is needed when step heating experiment is performed, as all the information are retrieved from the step heating (i.e., $^3$He$_c$ and $^3$He/$^4$He$_{mantle}$). Please justify the use of a crush value instead of the high-T value for mantle component.

**Table 4 and Table 5**: I found those two tables redundant. They show almost similar information, only isochron dataset is added to table 4. I would merge those two tables altogether in one clear table with the concentration and their associated calculated ages for all the methods used. It would be easier to have a table summarizing everything instead of two.

Reference:
**Blard** P.H., Balco G., Burnard P.G., Farley K.A., Fenton C.R., Friedrich R., Jull A.J.T., Niedermann S., Pik R., Schaefer .M., Scott E.M., Shuster D.L., Stuart F.M., Stute M., Tibari B., Winckler G. and Zimmermann L. (2015) An inter-laboratory comparison of cosmogenic 3He and radiogenic 4He in the Cronus-P pyroxene standard. Quaternary Geochronology, 26, 11-19.
**Kurz** M. (1986) In situ production of terrestrial cosmogenic helium and some application to geochronology. Geochimica et Cosmochimica Acta, 50, 2855-2862.
**Williams** A.J., Stuart F.M., Day S.J. and Phillips W.M. (2005). Using pyroxene microphenocrysts to determine cosmogenic 3He concentrations in old volcanic rocks; an example of landscape development in central Gran Canaria. Quaternary Science Reviews, 24, 211-222, 2005.
**Yokochi** Reika, Marty Bernard, Pik Raphaël, and Burnard Pete. (2005). High $^3$He/$^4$He ratios in peridotite xenoliths from SW Japan revisited: Evidence for cosmogenic $^3$He released by vacuum crushing. Geochemistry, Geophysics, Geosystems, 6, ff10.1029/2004GC000836f.
**Zimmermann** and Marty. (2014). Méthodes d'extraction des gaz rares sous ultravide. Techniques de l'ingénieur.
**Zimmermann** Laurent, Blard Pierre Henri, Burnard Pete, Medynski Sarah, Pik Raphael and Puchol Nicolas. (2012). A New Single Vacuum Furnace Design for Cosmogenic $^3$He Dating. Geostandards and Geoanalytical Research, 10.1111/j.1751-908X.2011.00145.x.
**Zimmermann** Laurent, Avice Guillaume, Blard Pierre-Henri, Marty Bernard, Füri Evelyn, and Burnard Peter. (2018). A new all-metal induction furnace for noble gas extraction. Chemical Geology, 10.1016/j.chemgeo.2017.09.018.

---

## Author Comment (AC1)

**Authors' Response to Referee Comments**

In black are referee comments
In red are author responses

Although probably covered by the blanks (re-extracts) done after each step heat temperature step. I just worry about the 3He/4He ratios of the lower T step heating steps possibly having a higher R because the 3He is successfully diffusing out of the matrix faster than 4He because of the difference in mass. Perhaps a little more discussion could be added on why this worry is likely unfounded? A little on diffusion rates of 3He and 4he with varying T. And/or consider publishing the re-extract data so readers can see that there was zero (or very low) 3He and 4He on the re-extract at 800C? This is important because the 800C step is the only one that really contributes 3Hec (as stated) by this step heat method.
To clarify, there were not re-extracts after each step heat temperature step. Three hot blanks were run on an empty furnace at the beginning of the experiment at each temperature step. Two re-extracts were run at the end of the experiment to ensure total release of He. I will add a citation and sentence to alleviate concerns about He diffusion.

This is not a huge point, but I'm not sure that the two-step method is always done as the authors suggested, namely that the step after crushing always involved completely powdered olivine/pyx sample. Often researchers just use the crushed leftovers (or sometimes uncrushed phenocrysts) and it works fine as there isn't much mantle gas, the crushing step opened most of the host inclusions and/or the exposure duration and site PR were high enough to overcome some left over mantle gas and get a high % 3Hec. Some labs will try using in vacuo crushed sample material first and then power if needed as the powdering step brings with it a host of problems as the author's state.
I added clarified that the second step does not always involve powdering.

Line by line comments:
18 – should give the temp step ranges of the three ranges used rather than just 700-1400C
I added in the temperature step ranges.

20 – 'youngest'…. how? Morphologically, relative age relationships?
I added in stratigraphically to describe what youngest means in the paper.

24 – I'd remove the 'for decades' to avoid confusion
I removed this part of the sentence for clarity.

35-37 – I'd list mantle He first and the other sources after that as mantle is the most important for young volcanics
I rearranged the sentence so that mantle He is listed first.

55-65 – I'd just mention that this newer variant of the isochron method was introduced in Blard 2021 and has different axes that the 'traditional' isochron method of Cerling and Craig, 1994; Blard and Pik , 2008; etc.
I clarified the change in axes.

66 – again, 'youngest' , how do you know they're the youngest?
I added in stratigraphically.

69 – not sure what 'following work on peridotites' means – has this step heat correction already been done and published before? Or is this saying that the peridotites analyzed here have already been studied which aided in developing this new method? Either way, perhaps rewrite this.
This sentence was rewritten for clarity.

75 – 14C ages are typically given as either 14C yr BP (if uncalibrated) or cal yr BP (if calibrated). The reference that is cited should tell which.
I added in yr.

88 – I'm not exactly sure what 'primary depositional features' are on a lava flow? Maybe primary lava flow morphology, or something like ropy texture suggests that it is a flow top with minimal weathering.
This sentence was changed for clarification.

101-105 – so the olivine samples were a mix of phenocryst and xenocryst olivine, as well as possible olivine from ultra-mafic xenoliths? Are they all seeing the same mantle gas environment? All the ways you calculate the 3Hem they look the same at least.
The olivine analyzed contains two types of olivine, the phenocrysts and xenocrysts. It is likely that the mantle gas trapped inside the olivine from the xenocrysts and phenocrysts ha

113 – multiple problems with exponents and units in this line
This is fixed now.

120 – is this a resistance furnace?
Yes, this is clarified now in text.

125 – second time with these exponents and units, maybe not a mistake but intentional. Describing a concentration as 10^-3 Matoms is a little confusing, similarly 10^3 M atoms is as well.
This is fixed now.

145 – so again with the exponent and M – why not just say 10^12 instead of 10^6 M…
[4]He concentrations are in atom/g now.

170 – the crush results in this table are not Matoms/g but rather Matoms. No leeway on this, you don't know how much mass was crushed.
We disagree. We know how much mass was placed into the crusher and how much was retrieved. These facts allow us to have a fairly accurate value for the mass and a crush concentration. This is a standard way to report [4]He concentration for crushes.
=
172-174 – this assumption is likely untrue and I'm not sure its 'curious' to test it in this way (which I'm ok with – testing it in this way that is…), but rather interesting, valuable, or illustrative?

Wording has been changed.

183 – the Thirumalai 2011 paper cited in the figure caption is not in the ref list.
This has been fixed.

185 – might be worth it here to remind readers that these step heat samples were powdered.
I will add a note about this in the figure caption. I think the methods section makes it clear that they are powdered.

205 – maybe before mantle in the end of the line put 'persistent', so its 'persistent mantle Helium'
I added in persistent.

237 – the 3 on 3He isn't superscripted
This has been fixed.

264 – not sure which PR calculator you're using, is it the FORMER Cronus Earth calculator, the Balco one…sometimes called the UW calculator, but not sure what it's routinely called now, I'd just put the web address for clarity.
Link added.

271 – for the average of the two ages, how is the uncertainty determined? If you average the ages the standard deviation might be better for the uncertainty (about 0.2). If you averaged the 3Hec and determine an age then seems ok to use the internal uncertainty associated with the exposure age calculator (more like 1.1 that is used). Should just say.
I have updated the manuscript to use standard deviation for uncertainty on the average age.

288—maybe remind readers of the association of TS peridotite to these samples? And you set up and used TS as shorthand for Twin Sisters earlier, maybe either use it consistently after that or just don't worry about shorthand for a location you only mention a few times.
The connection is that both samples have a source of mantle He that cannot be removed by powdering.

Figures
Fig 1 – I find some of the colors hard to match back to the legend, they are clearly labelled though.
We will leave this figure as is.

Fig 2 – maybe should put equation of that line inside the figure space rather than in the caption, and include statistical tests on that regression:  r2, p, MSWD etc. Should also note that the line is extrapolated beyond the two data clusters.
We will leave this figure as is. I have noted that the line is extrapolated beyond the two data clusters.

Fig 3 – this figure is key to the whole paper and is interesting. I would shorten the lines, to maybe like +/- 50 deg C around the T that you measured 3He and 4He at. As is, it looks like you

are declaring that there is some sort of degassing domain change at 900 C and 1200 C but I don't think you're really trying to say that?

This figure has been updated! The cause of the issue was a formatting error in Sigma Plot.

---

## Author Comment (AC2)

Here we respond to comments of reviewer P.-H. Blard

*This article provides an interesting method based on a sequential heating of olivines from surface lava flow samples. This technique permits to separate the cosmogenic 3He and the mantle 3He component. Their dataset obtained on 2 samples convincingly indicate that diffusion at 800°C preferentially release cosmogenic 3He, permitting to determine this cosmic-ray produced 3He with a better precision than using the standard crushing-melting method. This better precision is possible because the mantle component is not released at "low" temperature, reducing the impact of the uncertainty of the magmatic correction. However, although I think that this pilot study is interesting and should be published, I have major criticisms about the radicality of several conclusions of the authors, who write that the isochron method and the standard crushing-fusion methos yields imprecise or flawed ages. They cannot be so definitive and negative about these well-established methods, for two reasons: first, they didn't apply the heating step method on the same samples that those processed by the crushing-melting methods, and second, they build an isochron without using samples (or aliquots) that have been exposed to the same "dose" of cosmic rays. The most plausible conclusion is rather that their samples have various exposure ages (either because the lava have different ages, or because the sampled surfaces suffered differential erosion or shielding). I encourage authors to revise their manuscript taking into account this main criticism and also the other points listed below.*

The major point of the paper is to a) indicate that step heating can be used to reasonably isolate cosmogenic He from mantle He in these olivines that are extremely He retentive during crushing and b) to estimate the most recent eruption age. Blard here makes two legitimate points, but neither directly impacts the key results of the paper.

We agree that it is likely the samples we plotted in the isochron violate the assumptions we stated: "This same fusion data can be cast as an isochron provided we assume the flows have the same exposure age and mantle component." Regardless, in the end we agree that the isochron plot as presented was not useful or well described. In a revised submission we will modify the discussion of the isochron approach, instead focusing on how even if the fundamental assumptions of the isochron approach are absolutely true, the high mantle He concentration and its limited range in our powder fusion analyses coupled with reasonable analytical uncertainties inevitably yields substantial uncertainties on cosmogenic He concentration. In other words, we test the isochron's use under ideal conditions for our real-world samples, and assess the best-case precision that could be obtained.

We will also provide newly obtained crush results on the lava flow subjected to step heating.

*Line 40: ""low" is rather imprecise. Provide actual range of U and Th concentrations."*

*Line 42: This belief is not really accurate: young lava (< 100 ka) may require correcting for radiogenic 4He accumulation, because what matters is the 3Hecos/4Herad production ratio, that is independent from the lava age (e.g. Blard and Farley, 2008; Blard, 2021). Olivines that bear significant amount of U (> 0.1 ppm) require more than 10% correction (R factor lower than 0.9, see for example figure 9 in Blard, 2021). In other words, a "young" lava does not necessarily imply that the radiogenic 4He impact on the magmatic 3He correction is negligible (Dunai and Wijbrans, 2000 = 5% correction; Blard et al., 2006 = up to 12% correction, even in lavas younger than 200 ka ; see R factors in Table 2 in Blard and Farley, 2008).*

*Line 42: What is a "young" lava?*

All of this refers to our generic statements in the introduction that under certain limiting circumstances it is possible to reduce the nearly intractable 3 component problem (mantle + cosmogenic + radiogenic helium) to a solvable one that involves only the first two components. It would derail the logic of the introduction by digressing into the details requested here. Instead, we will add text to the discussion where we justify why, in these olivines with very high mantle concentrations and estimated eruption age of ~10 kyr, the limiting case is reasonably satisfied. The key conclusion is that there is so much mantle helium in these olivines that the radiogenic component in 10 ka olivines is not a factor of concern.

*Line 49: Maybe add a  caution here to mention that dry powdering under atmosphere may yield adsorption of non negligible amount of atmospheric helium on the phenocrysts/xenocrysts (Protin et al., 2016).*

Easily done in revised version.

*Line 51: Quote a source for the value of this atmospheric isotopic ratio.*

Easily done in revised version.

*Line 56: Figure 5 in Blard 2021 shows a modeling of the magmatic helium impact on the final 3He_cos uncertainty.*

We can point this out in revised version.

*Line 68: Note that this approach involving a step heating to selectively release cosmogenic 3He at around 1000°C is not a newly developed method. As written in Niedermann 2002: "The few papers which report stepwise heating data for cosmogenic He show the major release of 3Hec from mafic and ultramafic minerals (olivine, pyroxene) below ~ 900-1100°C (Kurz 1986a; Staudacher and Allègre 1991, 1993a; Sarda et al. 1993; Schäfer et al. 2000)."*

We are not claiming priority in demonstrating step heating He component resolution, we are just demonstrating its use for quantifying the cosmogenic 3He concentration in an unusually problematic set of samples. We can add these references in revised version.

*Line 75: Lava-dammed lakes are not shown on figure 1, could you please add them on this map? It would be very useful to clearly show the stratigraphic relationships between lavas and lakes.*

This can be done in revised version.

*Line 85: "Cosmogenic sampling". Since "cosmogenic" means "produced by cosmic rays", I suggest to reword.*

Easily modified in revision.

*Line 90: A picture of a sample would be very useful here.*

Easily modified in revision.

*Line 91: What is the typical sample thickness?*

Easily modified in revision.

*Line 97: It would be useful to show field pictures of the sampled lava surfaces (either in Figure 1, or in a new figure). Field photographs would be very useful to show the vegetation cover.*

We will mention the existence of field photos of these lava flows in published references, especially Jackson and Stevens 1992.

*Line 102-103: Are these mm-olivine phenocrysts or xenoliths?*

We analyzed olivine crystals that could be either ol phenocrysts or disaggregated xenoliths. The references we cite suggest that the majority of the olivines are the latter (disaggregated during eruption), as we already stated in the "Geologic Setting" section of the manuscript.

*Table 1: Why don't you provide thickness for samples 8, 10 and 11?*

Will be added in revision.

*Line 119: Along Cox et al 2022, quote here the first article that identified this issue of helium adsorption (Protin et al., 2016).*

Will be added in revision.

*Line 121: 1200 °C is well below the melting point of Mg-rich olivine (that is above 1500°C). Are you sure of 1) the accuracy of the furnace T control, and 2) are you really reaching fusion (isn't it an He extraction by complete diffusion)?*

In revision we will add text to the effect that we are using the term "powder fusion" to mean "complete extraction of He at high temperature". We verify complete He extraction using the re-extract method. We don't fuse the olivines because ~1800 C is beyond the range of our furnace.

*Line 135: Please provide the blank levels for each T step, as you did in the previous section for 1200°C.*

Will be added in revision.

*Line 136: It is suprising that the contribution of the 1400°C blank is lower than the one of the 1200°C blank.*

Why? The system is being cleaned up by the steps that occur earlier.

*Line 153: It would be useful to give the value of this shielding correction here.*

Corrections are listed in Table 5, but in revised version we can mention them here.

*Table 2: I think this is not necessary to provide 3He and 4He with 2 different units (cc/g and at/g). This would be much more useful and informative to add a column with the computed cosmogenic 3He concentration corrected for the magmatic 3He for each sample.*

Easily modified in revision.

*Line 172-173: "mantle component": I guess you mean "magmatic 3He/4He ratios", not magmatic helium concentrations, that need to be variable to ensure the isochron method working. Maybe rephrase to avoid confusion.*

Easily modified in revision.

*Line 176: Given that this isochron is built from samples having different exposure histories, I think you should use this number with a huge caution.*

Agree. The isochron section will be completely rewritten in revision, as described above.

*Figure 3: For a better readability, you could consider removing the vertical lines between each T extraction step.*

Figure will be revised.

*Line 209: Replace "u" by "mu", the greek m.*

Easily modified in revision.

*Line 218-219: Burnard et al., 2015 also provided experimental evidence for the presence of a significant "reservoir" of mantle helium at the grain boundaries.*

There are no grain boundaries in our analyzed olivines- we are analyzing individual crystals. We can mention in revision that we have crystals, not small polycrystalline xenoliths as this comment seems to be imagining.

*Figure 4: It would be useful to add an enlarged zoom centered on these "trail" of fluid inclusions.*

Figure will be revised.

*Line 237 and 240: It would be useful to add these relative increases in cosmogenic 3He in Tables 2 and 3.*

We will do this.

*Line 240: "34 and 39%": Why are these 2 numbers different from the 36-42% range mentioned line 237?*

This will be clarified in revision.

*Line 241: What do you mean by "detailed step heat"? Do you mean an initial heating at lower tempertature?*

It is not obvious what temperature the additional steps would need to be to obtain best component resolution. This will be clarified in revision.

*Line 253: It is strange to compute a mean using a set of samples with heterogeneous ages.*

In the end we agree this mean computation adds no value. This section will be reworked in revision.

*Line 266: "7.7 +- 4 ka". I think there are too many significant numbers.*

Easily fixed in revision.

*Line 269 to 271: You should rather compare VM1 and 3 with VM2, 6 and 8, not with the isochron that is built from a dataset that breaks the required assumption of an homogeneous dataset.*

*Line 279: I disagree with this conclusion. You could conclude that if the same 3 methods were applied on the same samples. Here you applied different methods on different surface samples. The most plausible conclusions from your data is that some samples (9, 10, 11) have experienced less exposure at the surface than others (1, 2, 3, 6 and 8). Regarding the building of the isochron, you did not respect the necessary assumption of using aliquots with homogeneous cosmogenic 3He concentrations, so you obtain an underestimate average concentration. Nothing else can be concluded from that.*

*Line 291-292: As already mentioned before, this conclusion is a false overstatement. You cannot provide such a general conclusion about the isochron and the crush-fusion methods since you did not apply them to the same samples.*

As already mentioned, we will rework the discussion of the isochron and add a crush measurement in revision. These will address this entire set of concerns.

---

## Author Comment (AC3)

Response to review by J. Amalberti

The fundamental observation made in our manuscript is that Volcano Mountain olivines have a mantle He component that is not effectively removed by the crushing technique used most commonly to isolate mantle from cosmogenic 3He. The inability to remove the mantle component of 3He by crushing made it difficult or impossible to obtain a reliable estimate of cosmogenic 3He in our samples using the standard crush/fusion approach. Nor does an isochron approach yield reliable results for these samples. As a consequence, we investigated a step heating technique as an alternative, and found it satisfactory for estimating the age of the most recent VM flow. So, in addition to the most recent eruption age estimate, the key point of the paper is that in these unusual samples with high concentrations of mantle helium that survive crushing to powder, step heating provides an alternative approach for cosmogenic 3He dating.

**Step Heating Component Resolution**

Amalberti provides an extremely lengthy review of the manuscript. Specific comments are addressed in detail below, but Amalberti has one central issue that he claims challenges the basic premise of the step heating approach we are demonstrating in this paper. He is concerned that step heating is not separating matrix-hosted cosmogenic He from fluid-inclusion-hosted mantle He, but is instead separating two different mantle components, an OIB-like mantle component released "from small inclusions" at low temperature, and a MORB-like component released from larger inclusions at higher temperatures. We reject this alternative for the following reasons:

1) To our knowledge such a heterogeneously distributed mantle component has never previously been observed in a population of olivines from a single lava flow. Why is VM special in this regard? Moreover, it is difficult to accept that He isotopic heterogeneity could be maintained within olivine crystals at magmatic temperatures. While one could arbitrarily construct a story of a magma in which the 3He/4He ratio is changing and these changes are passively recorded as the olivines trap fluid inclusions, the diffusivity of He at magmatic temperatures is sufficiently high that He would be expected to equilibrate isotopically over the mm scale of individual olivines, and even among a population of such olivines (see discussion in Horton et al, 2019 and the aside in the next paragraph).

2) Amalberti justifies his proposed resolution of two mantle components via step heating by appealing to the novel idea that small fluid inclusions will have He diffuse out faster than large inclusions and will thus be sampled preferentially in the low temperature steps. This statement does not follow from the relevant physical phenomena. The process of diffusion is the same regardless of inclusion size: He partitions into the olivine matrix from the inclusion, then diffuses through the matrix and out of the grain. Diffusivity is not faster because the He was delivered to the matrix from a smaller inclusion. (As an aside, if He in these hypothetical small inclusions diffuses out of the olivines on an hour time scale in our experiments at 800 C as Amalberti suggests, then surely these bubbles would equilibrate He internally to the olivine at magmatic temperatures, as noted in item 1, above. The "two mantle component hypothesis" is not even self-consistent).

3) In contrast to multiple mantle components, isolation of a cosmogenic component at low temperature and a fluid inclusion component at high temperature has already been compellingly observed, e.g., by Swindle et al (2023), who found that 3He/4He ratios in excess of 150 Ra can be extracted at low temperatures, but mantle-like 3He/4He ratios around 5 Ra at high temperature, in cosmic--ray exposed olivines from the Twin Sisters massif. The revised manuscript will provide additional references to this effect. This is readily understood based on known siting of these gases. Cosmogenic He is confined to the olivine matrix which has relatively high He diffusivity. In contrast, He in fluid inclusions must partition out of fluid inclusions before diffusing - impeding its thermal extraction. See discussions by Blard et al 2008, Trull and Kurz, 1993, and Horton et al, 2019.

We thus stand behind our preferred interpretation that step heating is isolating a cosmogenic component from a mantle component, and the validity of our reported VM eruption age. In a revised manuscript we will include new olivine crush measurements on the samples we step heated that directly support this conclusion.

**Isochron Plot**

Amalberti takes issue with our presentation of the isochron plot objecting (again) to our assumption of a single mantle component.  Obviously an isochron requires both homogeneity of the mantle component and the cosmogenic concentration. This is why our manuscript states:

"This same fusion data can be cast as an isochron provided we assume the flows have the same exposure age and mantle component."

Regardless, in the end we agree that the isochron plot does not usefully constrain the eruption age (see response to review by Blard). In a revised submission we will modify the discussion of the isochron approach instead focusing on how even if the fundamental

assumptions of the isochron approach are absolutely true, the high mantle He concentration and its limited range in our powder fusion analyses coupled with reasonable analytical uncertainties inevitably yields substantial uncertainties on cosmogenic He concentration. In other words, we test its use under ideal conditions, and assess the best-case precision that could be obtained.

Below we list Amalberti's specific comments highlighted in gray, with our response unhighlighted.

The article seems to point out that they developed a new methodology to analyze 3He cosmogenic signal by step heating the samples (see lines 68-69, and 292). This method, however, is not recent and has been used and developed by other authors before (Kurz 1986a). It would be more appropriate to refer at the step heating method as an already known approach (such as the crush/fusion or the isochron) but mentioning that you have adapted it with three temperature steps (800, 1000 and 1400°C).

This comment is ungenerous. Kurz 1986a did not use a step heat to quantify the cosmogenic component in an olivine sample with extremely retentive mantle component.

Line 20: "precise estimates of cosmogenic" is not really supported by the original manuscript as the step heating experiment (where the 3Hec is the best measured) shows only one low temperature step, which already limit the precision of the 3Hec determination. Indeed, very low 4He* contribution (Kurz 1986a) or a variable mantle component cannot be investigating with such low-resolution step heating, and could likely affect the 3He/ 4He ratio.

This statement is exaggerated. As described in the text VM olivines are very likely disaggregated mantle xenoliths. Such xenoliths have extremely low U and Th concentrations, likely much less than 0.1 ppm (e.g., see measurements of Twin Sisters peridotites in Swindle et al 2023). This amount of U would generate less than 0.5% of the 4He we measure in our low temperature steps. This would make a tiny difference to the correction for mantle helium. We also reject the idea of multiple mantle components in the step heated olivines for reasons already mentioned. Of course we could do more experiments with more temperature steps, but the results would not meaningfully change our conclusions.

Line 66: I disagree here. The isochron method was not used in the paper, otherwise aliquot from same olivine population would have been used. In addition, the equation for the isochron given by Blard and Pik (2008) is not used here. The authors simply plot 4He vs. 3He following equation 5 in the text which correspond to the classic crush/fusion method and called that an isochron. I would either remove isochron method statement, or if you

want to assume that the 6 lava flows have same age, and can be used as one population, it should be clearly stated and the proper isochron equation should be used to verify the validity of the method. Please see my major comment above for more detailed discussion on the crush/fusion and isochron dataset.

This is a very frustrating comment. We stated *exactly* what the reviewer states about assumptions on line 172: " This same fusion data can be cast as an isochron provided we assume the flows have the same exposure age and mantle component."

The statement that we did not use an isochron is wrong. The form of the isochron we used was shown in Blard (2008) and follows from exactly the same set of assumptions and equations as the form preferred by this reviewer.  On what grounds is this reviewer claiming we did not use an isochron method? Our approach is NOT just the traditional crush and fuse method, utterly evident from the fact that the crush data never enters into the plotted data!

Line 68: "we developed" should be replaced by "we used" or similar phrase.

Yes, this ungenerous point was made earlier.

Line 91-92: The authors specified that they have additional notes on the samples, such as vegetation cover or average sample depth, but failed to provide those data (or I didn't see them). I would have like to get them in a supplementary material, specially that some detailed sample information could have be beneficial.

The sample depths are shown in Table 1, apparently missed by the reviewer. We do not believe the effects of vegetation cover are significant especially at the level of (im)precision we can obtain, so have not included these data in the Table.

Line 102-103: The authors specify in this section that 4 samples are disregarded and 8 samples are selected due to the mm-sized olivine. Could you please specify if no mm-size olivine were found in those 4 lava samples or if the olivine quality was insufficient for proper 3Hec investigation? It is unclear from the description why those samples are disregarded. In addition, where is sample VM-07? It is supposed to be used in the study (see line 96 and line 103) but I cannot find this data in the text, the tables or the plots.

This question was clearly addressed in the text:  "VM-01-3, 06, and 08 through 11 (Table 1) were chosen for this study due to their higher modal abundance of mm-sized olivine grains (~5%) compared to samples VM 04, 05, 12, and 13". We concentrated on samples where

we could pick sufficient olivine for this study. VM 07 was inadvertently left off the list of low modal ol abudances.

Table 1: This table could be more interesting if more information regarding the sample notes were included such as the vegetation cover or the topography shielding.

The shielding correction is shown in Table 5, which includes the other data used to calculate exposure ages.

Line 111: The authors specify that the crushing protocol is derivated from Blard et al., 2008. There is no Blard et al., 2008 in the reference list. I found however, Blard and Pik (Chemi. Geol. 2008) and Blard and Farley (EPSL 2008). Please be careful when referencing literature (see comment at line 182 as well). In any case, both Blard and Pik (Chemi. Geol. 2008) or Blard and Farley (EPSL 2008) papers do not contain any indication (or I couldn't find any) for a crushing step at 2min followed by a step at 5min for proper 3Hec extraction. Could you please provide information where this protocol has been taken? In addition, early crushing steps can potentially release 3Hec from the matrix (see Blard and Pik (Chemi. Geol. 2008) and Blard and Farley (EPSL 2008) papers for more details). Do you have investigated such 3Hec loss/contamination on the 3He/4He crush signal? This could lead to important impact on 3Hec measurements if not well estimated. It should be at least mentioned here.

We will correct the reference to the crusher we used (Blard, Puchol and Farley 2008). We did not do a study of how much cosmogenic 3He might be released by this device, but it is the same device described in Blard et al (2008), where the release of matrix-sited He as a function of crush time is shown. There is no observed release for the crush durations we used.

Line 121-122: Please add the re-extraction data to the dataset (in the main text or in supplementary). I noticed also that fusion is performed at ~1200ºC for 25min but step heating experiments show un-degassed samples at T~1400ºC for 30min step (last temperature step at table 3). I would recommend therefore to provide all the re‑extract dataset to ensure that the fusion and step heating samples have been properly outgassed. At the moment, it seems difficult to fully outgas the samples with one temperature step at ~1200C for 25min. If re-extraction shows significant He, did you then add them to the total? Please provide additional information.

We will add text stating that re-extracts were repeated until no He above blank levels was obtained, and the reported concentrations reflect the sum of all steps. The reviewer is mistaken about the 1400 C. This step integrates any helium that was released by the sample between 1000 and 1400 C. It says nothing about retention above 1200 C. We have certainly degassed the entire sample.

Line 124-125: The authors are using a blank correction for the He analysis with an empty furnace while using Sn foils to wrap their samples. I would have expected blanks to be run with empty Sn-foil packet instead, to better account the blank value. Same comment can be given for step heating experiment (see line 135). Given the blank level is given at 5% (without Sn-foil contribution), I suspect that blank could be underestimated if He outgassing for the Sn foil is not accounted for. In addition, 5% He blank contribution is not insignificant to my opinion.

Sn foil carries no substantial additional blank above that which came out of the hot furnace in this set of experiments. We will add text to this effect and eliminate use of the word "insignificant".

Line 134-135: Please provide the re-extract data.

Same comment as above.

Line 157: What means early measurement here?

Sentence removed - it was irrelevant.

Line 166: Using equation 5, for sample VM-08, I calculate a 3Hec at 2.8 Matoms/g instead of 2.58 Matoms provided by the authors. I am using an average 3He/4Hecrush at 1.12x10-5 (from VM-06 and VM-09). I suspect that you are either using VM-06 or VM-09 crush value but without any justification. Therefore, for the sake of consistency, I would recommend to use VM-06 and VM-09 average crush value for all VM samples at the exception of VM-06 and VM-09 where 3Hec can be properly determined. Otherwise please justify which crush values you are using for all VM analyzed samples.

Strangely, this reviewer is proposing exactly what we did. Apparently the reviewer missed our text that reads: "Given that the crush ratio of 8.1 ± 0.2 $R_A$ is likely representative of all VM samples, there are six crush-fusion pairs that can be used to calculate $^3He_c$ concentrations using Eq. 5.".

We will clarify the sentence to make it more noticeable.

Table 2: For clarity purpose, I would add a special label for the two samples that have coupled crush/fusion data. The other could be labelled as uncrushed olivine.

Good idea.

Fig. 2: Please give the linear regression value in the plot. This is important information and should be display clearly in the plot, not in the caption. Also, as mentioned in my major comments above, the data plotted here is not representing the data in table 2. Please check why this is not the same as this might lead to different 3Hec given by this regression line based from equation 5.

There was a plotting error that will be fixed in the revised manuscript. We can add the numerical results of the regression to the plot since this reviewer was unsatisfied by its location in the caption.

Line 182: Thirumalai et al., (2011) and York et al., (2004) are not listed in the reference section. Please be sure all the references are included.

Good catch. Easily fixed.

Fig. 3: The red solid line for 3Hec is likely to be wrong. The calculated 3Hec from equation 5 using VM-1 data at T~800ºC lead to 3Hec calculated at 3.1 Matoms/g (which represent 92% of the total 3Hec for VM-01), but in the figure, the red solid line is showing a value >5 Matoms/g. The black solid line seems ok, but could you please check if the plot has the proper values calculated for 3Hec ? In addition, the last temperature step (T~1400ºC) is very likely not fully outgassed. The signal is still showing high He content. If you have the re-extraction, please provide them to ensure that total outgassing of the sample is performed.

This plot will be redrafted because it is indeed incorrect. We have already refuted this reviewer's belief that we have not completely extracted He from these olivines.

Lines 240-241: I think the argument for more detailed step heating is important here. Knowing that early temperature step can contain important information, especially for 4He* contribution that can lead to a lower Ra value at low T-step (see Kurz 186a), or if we suspect some 3He/4He lower mantle contribution from the tiny inclusions. I would have

therefore, expected a better resolution for the step heating experiments. The SFT is capable of analyzing significantly lower values of 3He, much lower than ~0.4 pcc/g (3He concentration given at T~800°C), especially knowing that blanks value is given at ~0.8 fcc, and therefore, in the worst-case scenario, acceptable values for 3He could be potentially measured as low as 0.008 pcc (~10 times the blank). However, as a theoretical example, if a lower temperature step (T~500°C) is performed on sample VM-01, and if 3He signal is measured 100 times lower than the one at T~800°C (i.e. ~0.07 Matoms), therefore the blank contribution given at 0.002 Matoms should "only" represent ~3% of the signal, which is quite acceptable. 4He signal, on the other hand is more complicated and could lead to some limitation under the current analytical blank of the double wall furnace. The 4He blank is given at 0.2 ncc which limit the measured signal at ~2 ncc of 4He (~10 times the blank to ensure sufficient precision). It is noteworthy that signal lower than 10 times the blank can be measure but then larger error is expected and could limit the interpretation. Nevertheless, 4He signal 10 times lower than the measured ones at T~800°C (i.e. $4.3 \times 10^4$ Matoms) could still be measured for a hypothetical step at T~500°C. Blank error could account for ~12% (5380 Matoms). I would like, however, to point out that the double-wall furnace (where step heating experiment is performed) is not baked but solely pumped out overnight (see lines 120-121), and 4He blank given by the authors are quite high compared to some double wall furnace blanks given by other studies (Blard et al., 2015; Kurz, 1986; Williams et al., 2005, Yokochi et al., 2005, Zimmermann et al., 2012; 2018, Zimmermann and Marty, 2014), where blank values are given as low as ~300-600 Matoms. If we assume that blank values can be reduced significantly in the range of 0.02 ncc (~600 Matoms), then previous blank contribution of ~12% (for a hypothetical T~500°C step) will drop at ~1.5%, and 4He concentration 100 times lower could be even analyzed. In addition, peak jumping analyses can also be performed to measure 4He on the CDD to keep good 4He measurement precision for very low signals. Consequently, I think the authors could safely performed a more detailed step heating experiment with limited loss of the analy

This reviewer is proposing acquisition of additional step heat data. We decline as it would require substantial additional work and in our opinion would add nothing new. We have already offered our refutation to both of this reviewers motivations for this additional work.

Line 245-247: Why the authors are using the crush value here, while they have the mantle component determined with the highest temperature (T~1400°C, where no cosmogenic contribution is estimated, see line 249)? I would rather use the high temperature 3He/4He ratio for the mantle composition instead of using a crush value from another lava flow (VM-

06, VM-09 or average). My understanding is that no crush step is needed when step heating experiment is performed, as all the information are retrieved from the step heating (i.e., $3He_c$ and $3He/4He_{mantle}$). Please justify the use of a crush value instead of the high-T value for mantle component.

The revised manuscript will include a crush analysis on the olivine from this flow, eliminating this concern.

Table 4 and Table 5: I found those two tables redundant. They show almost similar information, only isochron dataset is added to table 4. I would merge those two tables altogether in one clear table with the concentration and their associated calculated ages for all the methods used. It would be easier to have a table summarizing everything instead of two.

Agree. Tables will be streamlined in revised version.

---

## Author Response (AR1)

**Editor Notes and Revisions**

From Dr. Greg Balco

First, reviewers took particular exception to the use of an isochron method in a way that might be proper (if in fact all the samples from different locations have the same exposure history) or might be improper (if they don't). While it is true that the paper as written clearly states that the samples are from different locations and might not have the same exposure history, I agree with the reviewers that this is an improper application of the isochron method -- it might be right in this case by accident, but in fact the sample set doesn't strictly satisfy the assumptions needed to apply an isochron method. Thus, I strongly suggest removing discussion of the isochron approach from the paper. This would also simplify the paper somewhat, which would be helpful.

**We agree with this assessment and have removed it from the paper. Also, you will notice in the tracked changes file that some of the citations are highlighted in yellow. I had to edit these without tracked changes turned on because Zotero wouldn't work with it on. You will see in the new submission the citations are corrected.**

Second, while, again, it is true that the paper as written does contain all the information about which samples were subjected to which analyses, reviewers had numerous comments that indicated that they were unclear as to when/whether observations made on one sample could/should be applied to another sample. Please try to be more clear about this in your revision, possibly by stepping through your reasoning more explicitly - '...if this observation for sample A also holds for sample B, then...'

**We have added in and changed multiple sentences in the document so that the reasoning is clearer. You will also notice that we conducted another step-heating on VM-10 so that we could have both the stratigraphically youngest and oldest samples represented in the step-heating experiments. With this added information and combining both the C/F and step heating data, we determined that the four Volcano Mountain lava flows erupted approximately coevally, at 10.5 ± 1.7 ka. Because of this and from the feedback received, we have created a new version of our step-heat figure that is easier to follow.**

Third, in your revision try to be clear about the generality, or lack thereof, of the applicability of the method. Do you think that step-heating is likely to separate helium components only in this unusual lithology, or in others as well? Is there any evidence from the literature as to whether this should or shouldn't work in any particular situation? Note that I am not asking for a lot of speculation about where various helium

components are physically sited, etc., etc., but mainly just for a concise and realistic assessment of if/when it might be worth pursuing this in other samples/lithologies.

**We have adjusted our discussion section to include the elements you are referring to.**

With regard to the more specific review comments, these reviews included a large number of technical comments that, as you say, can mostly be addressed by fairly simple clarifications of the text. There are also a lot of comments relating to the history of various analytical methods, who did what first, and which papers should most properly be cited for various technical points. Please try to correct these issues as the reviewers suggest, but please also keep in mind that this is not a review paper and it is not necessary to summarize the entire history of trying to deconvolve various helium components for purposes of exposure-dating. It would probably be appropriate to give a short description of the various methods proposed and refer the reader to review papers for the details.

**We have added in the proper citations and more sentences to give credit properly.**

One technical comment I will address specifically is the suggestion that the results may reflect mixing of two mantle/inherited components rather than one mantle component and one cosmogenic component. Although of course it is difficult to completely exclude this possibility without shielded samples of this lithology, I agree with your reasoning here that (i) there is not an obvious mechanism by which two separate helium inventories could be kept distinct during the likely thermal history of the rock, and (ii) because the samples are now at the surface, we know there must be a nonzero cosmogenic component.

Finally, I am appending here four technical comments from the fourth person who supplied me with comments after the online review period closed; please also take these into account in your revision, especially the remarks related to Table 4.

1. The paper implies throughout that it is possible to completely separate the cosmogenic and mantle helium components in olivine by crushing. Mantle helium dominates in the inclusions, but mantle and cosmogenic helium are mixed in the olivine itself. The text will be misleading to non-expert readers in a few places. A few examples: Line 56 "When powdering does not effectively remove the mantle component". In my experience, powdering never fully removes the mantle component; Line 206: "the greater complication arises from the fact that mantle helium is not effectively removed by powdering to < 30" This is true, but is expected; Line 217 "

Survival of the mantle component when crushing this fine is not typical" This statement is probably incorrect. I know of no examples where crushing completely removes the mantle component, for xenoliths or basaltic crystals.

The original method of coupled crushing/heating never assumed that the two components could be separated, but that they can be "distinguished" by virtue of different residence sites and drastically different isotopic compositions. Much of the mantle helium is held in melt and fluid inclusions (easily released by crushing, which does not release much cosmogenic helium) and most of the cosmogenic helium resides in the solid mineral matrix. I cannot think of any examples where they are fully separated in heating measurements, so the text should be edited to reflect this. There are examples where cosmogenic helium dominates due to long exposure. If I am wrong here, then the text should include references to support the assertions.

**We adjusted this sentence and clarified this in the paper. We agree that separated isn't the best wording.**

2. One unique aspect of this sample suite is that the xenolith olivines have very high initial mantle helium concentrations and the lava flows have fairly short exposure ages of ~ 10Ka. The samples are apparently all xenolith olivines. Most of the basaltic cosmogenic helium data in the literature comes from olivines that grew in a basaltic melt, rather than xenoliths, and have much lower helium contents typically < 10 ncc $^4$He STP/gram. Therefore, this study may be a special case in the application of cosmogenic helium, leading to higher detection limits to the abundance of mantle helium. This should be pointed out somewhere in the text.

**I have added this into the text, thank you for the suggestion.**

3. Table 4 is misleading and should be modified, along with the text discussing it. The table gives cosmogenic 3He calculations for VM-09, 10, and 11. However, the fusion measurements for all three of these samples are indistinguishable from the crushing measurements and therefore the cosmogenic 3He is actually below detection, so tabulating cosmogenic 3He contents here is misleading, since it is below detection. It would be better to give the "upper limits" for those samples based on some estimate of detection limits. One simple way to calculate this would be to estimate lowest possible 3He/4He that would be distinguishable from the crushing data (e.g., 3He/4He of two or three standard deviations above crushing) and use that to estimate detection limits for cosmogenic 3He. (However, other factors like reproducibility and variable blanks may play a role too, so this is up to the authors.) If these detection limits are below the other three samples, is there some plausible geological explanation, such as erosion or soil or snow cover? Or perhaps these three samples were collected from a younger flow or

flows? Given the fact that the cosmogenic 3He could not be detected, it is not justifiable to include those three "zero-age" samples in combination of the other three that yielded more reliable measurements (the average of the six samples, line 253). It would be better to take the average of the three samples for which it was possible to detect 3Hec, which would lead to a different conclusion, i.e. that the crush/fusion results are significantly higher than the isochron method. At least discuss this possibility.

**We entirely agree with this assessment of Table 4 and have changed the format. Most of the tables have been updated to be easier to follow and understand.**

4. Figure 1 caption should include a reference for the map (Jackson and Stevens, 1992).

**We have fixed this.**

Overall, the result of all these reviews is a fairly extensive set of revision instructions. I hope what you will take from this is that the large number of reviews indicates that the paper is interesting and potentially valuable to readers, and therefore I hope that you will undertake these revisions.

-- greg

---

## Author Response (AR2)

As requested we converted concentration units to atoms/g (3He) and pmol/g (4He). We also updated Figure 2 to improve readability.